# [^89^Zr]-Atezolizumab-PET Imaging Reveals Longitudinal Alterations in PDL1 during Therapy in TNBC Preclinical Models

**DOI:** 10.3390/cancers15102708

**Published:** 2023-05-11

**Authors:** Adriana V. F. Massicano, Patrick N. Song, Ameer Mansur, Sharon L. White, Anna G. Sorace, Suzanne E. Lapi

**Affiliations:** 1Department of Radiology, The University of Alabama at Birmingham, Birmingham, AL 35233, USA; 2Department of Graduate Biomedical Sciences, The University of Alabama at Birmingham, Birmingham, AL 35233, USA; 3Department of Biomedical Engineering, The University of Alabama at Birmingham, Birmingham, AL 35233, USA; 4O’Neal Comprehensive Cancer Center, The University of Alabama at Birmingham, Birmingham, AL 35233, USA; 5Department of Chemistry, The University of Alabama at Birmingham, Birmingham, AL 35233, USA

**Keywords:** immuno-PET, ^89^Zr, [^89^Zr]-Atezolizumab, PDX, dosimetry, MDA-MB-231

## Abstract

**Simple Summary:**

Triple-negative breast cancer is characterized by a lack of targetable treatment receptors and current standard of care options, such as radiation therapy and chemotherapy, are associated with acute patient toxicity. Noninvasive imaging of PD-L1, with [^89^Zr]-Atezolizumab-PET imaging, has the potential to identify tumors with increased susceptibility to immunotherapy. In this work, we have optimized the labeling conditions of [^89^Zr]-Atezolizumab and used noninvasive [^89^Zr]-Atezolizumab PET imaging to characterize the longitudinal changes in PD-L1 expression in TNBC treated with standard of care treatment options. The goal of this study is to understand how [^89^Zr]-Atezolizumab PET imaging can characterize changes in intratumoral molecular biology and how this can inform eventual response and combination therapy.

**Abstract:**

Triple-negative breast cancers (TNBCs) currently have limited treatment options; however, PD-L1 is an indicator of susceptibility to immunotherapy. Currently, assessment of PD-L1 is limited to biopsy samples. These limitations may be overcome with molecular imaging. In this work, we describe chemistry development and optimization, in vitro, in vivo, and dosimetry of [^89^Zr]-Atezolizumab for PD-L1 imaging. Atezolizumab was conjugated to DFO and radiolabeled with ^89^Zr. Tumor uptake and heterogeneity in TNBC xenograft and patient-derived xenograft (PDX) mouse models were quantified following [^89^Zr]-Atezolizumab-PET imaging. PD-L1 expression in TNBC PDX models undergoing therapy and immunohistochemistry (IHC) was used to validate imaging. SUV from PET imaging was quantified and used to identify heterogeneity. PET/CT imaging using [^89^Zr]-Atezolizumab identified a significant increase in tumor:muscle SUV_mean_ 1 and 4 days after niraparib therapy and revealed an increased trend in PD-L1 expression following other cytotoxic therapies. A preliminary dosimetry study indicated the organs that will receive a higher dose are the spleen, adrenals, kidneys, and liver. [^89^Zr]-Atezolizumab PET/CT imaging reveals potential for the noninvasive detection of PD-L1-positive TNBC tumors and allows for quantitative and longitudinal assessment. This has potential significance for understanding tumor heterogeneity and monitoring early expression changes in PD-L1 induced by therapy.

## 1. Introduction

Triple-negative breast cancer (TNBC) is a subtype of breast cancer characterized by the absence of targetable growth receptors. TNBC represents 15–20% of breast carcinomas and is more prevalent in women under 40 years old, African American, Hispanic, and Indian women. TNBC is the most aggressive breast cancer subtype and is associated with poor prognosis, high metastatic potential, and limited treatment options [1,2,3]. TNBC lacks receptors for estrogen, progesterone, and HER2/neu, and, as such, it does not respond to therapies targeting these receptors. The standard-of-care for TNBC typically involves chemotherapy with drugs such as anthracyclines (e.g., doxorubicin) and taxanes (e.g., paclitaxel), and there has been a recent increase in the use of immunotherapy with checkpoint inhibitors as a promising treatment option [2,4,5]. The PD-1/PD-L1 checkpoint pathway acts directly on T-cell activation, proliferation, and cytotoxic secretion [6]. The activation of PD-1/PD-L1 signaling serves as a mechanism to evade T-cell immunologic response [2,4]. The primary goal of checkpoint inhibitors, such as atezolizumab (a humanized monoclonal antibody against PD-L1), is to induce tumor regression by reactivating anti-tumor immune responses [7]. In TNBC, atezolizumab demonstrated substantial overall survival in patients with PD-L1-positive (PD-L1+) metastatic or inoperable locally advanced TNBC [8]. Currently, PD-L1 expression is confirmed by immunohistochemistry (IHC), which can have low accuracy due to its sampling size: the tissue collected represents only a small site that may not accurately represent the intra and inter-heterogeneity of PD-L1 expression across multiple tumor sites or how it temporally changes during other therapies [9,10,11,12]. This can explain why patients with high tumor PD-L1 expression may not benefit from checkpoint inhibitor therapy, or patients who exhibit lower levels of PD-L1 expression may demonstrate a more favorable response than initially anticipated, considering IHC [13], thus making it evident that a better diagnostic tool is needed to select for PD-L1 therapy.

Molecular imaging is a noninvasive tool that provides spatial and temporal information on tissue-targeting and target expression. Positron Emission Tomography (PET) imaging can assess the target expression in all tumor foci to diagnose, stage, monitor, or treat several types of cancer. This has enhanced the field of image-guided therapy including studies incorporating imaging with PET agents to study or predict responses to immunotherapy. Examples include [^89^Zr]Df-IAB22M2C (a mini-body against CD8), which is being used to measure T-cell infiltration [14], and [^68^Ga]GZP (a Granzyme B probe), which can be used as an indicator of T-cell activation [15]. Longitudinal assessment of PD-L1 expression may provide the opportunity to both guide patient selection as well as provide a noninvasive whole-body assay to complement immunohistochemistry [9]. Preclinically, several authors have investigated anti-PD-L1 antibodies, fragments, and mini-bodies for PET/CT imaging [5,9,11,12,16,17,18,19,20]. Clinically, Bensch F. et al. [13] used atezolizumab radiolabeled with zirconium-89 (^89^Zr) to perform the first-in-human assessment of [^89^Zr]-Atezolizumab, and they found that imaging signal corresponds to PD-L1 expression. In addition, they also found the high tracer uptake seemed to be a strong predictor of response to atezolizumab treatment. These encouraging results make PD-L1 a promising target to be explored in molecular imaging, since it may be used for the initial disease staging, prediction of clinical outcome, monitoring of the therapeutic efficacy, and selection of therapeutic options.

In this work, we describe the chemistry development and optimization of Atezolizumab radiolabeled with Zr-89, which has a favorable half-life (78.4 h) for radiolabeling antibodies since it matches the time antibodies require for tumor accumulation, resulting in an optimal tumor-to-background signal [21]. We characterized [^89^Zr]-Atezolizumab in vitro and in vivo using imaging studies and demonstrated its potential to assess changes in PD-L1 expression following cytotoxic treatment in patient-derived xenografts, which better recapitulates the clinical heterogeneity of the disease and is a clinically relevant model of breast cancer. This work provides the chemistry and groundwork to translate [^89^Zr]-Atezolizumab into clinical trials.

## 2. Materials and Methods

### 2.1. Reagents and Materials

[^89^Zr]oxalate was produced at the Cyclotron Facility at the University of Alabama at Birmingham (UAB) from the ^89^Y(p,n)^89^Zr reaction using ^89^Y sputtered targets as described previously [22]. Atezolizumab (Tecentriq^®^) was purchased from Genentech (San Francisco, CA, USA). Desferrioxamine-p-benzyl-isothiocyanate (DFO-Bz-NCS) was purchased from Macrocyclics (Dallas, TX, USA). All other chemicals were purchased from Fisher Scientific (Hampton, NH, USA) except where otherwise stated.

### 2.2. Tissue Culture

Breast cancer cell lines MDA-MB-231 and HCC38 were purchased from American Type Culture Collection (ATCC, Manassas, VA, USA) and cultivated in Dulbecco’s Modified Eagle Medium (DMEM) and Roswell Park Memorial Institute 1640 (RPMI), respectively, with 10% FBS and gentamycin (50 mg/mL) in a humidified incubator with 5% CO_2_ at 37 °C. All other reagents for cell culture were purchased from Gibco^®^ Life Technologies (Grand Island, NY, USA).

### 2.3. [^89^Zr]-Atezolizumab Synthesis and In Vitro Characterization

#### 2.3.1. Synthesis of [^89^Zr]-Atezolizumab

Atezolizumab was conjugated to DFO-Bz-NCS and labeled with [^89^Zr]oxalate according to previously reported methods [23,24,25]. Briefly, 2, 5, 10, and 20-fold molar excess of DFO-Bz-NCS (8 mg/mL, diluted in DMSO) was conjugated to atezolizumab (1 mg; 17 µL) in 0.1 M sodium carbonate buffer pH 9 (q.s. to make total volume of 100 µL) at 37 °C for 1 h. After conjugation, DFO:atezolizumab was purified twice via Zeba spin desalting columns using 1 M HEPES buffer pH 7.1–7.3 as an eluent to remove the excess DFO. The final concentration of protein was determined using a bicinchoninic acid (BCA) assay (Thermo Scientific, Rockford, IL, USA). The purified conjugated antibody was radiolabeled with neutralized [^89^Zr]oxalate (0.15–0.74 MBq/µg) at 37 °C for 1 h and radiochemical purity was determined by instant thin-layer chromatography (iTLC) using 50 mM DTPA as the developing solution. The stability was tested by incubating it for 24 h at 2–8 °C, which leverages other work demonstrating long-term stability of radiolabeled antibodies [23,26]. At the end of the incubation period, the radiochemical purity was tested by iTLC as described (Appendix A).

#### 2.3.2. In Vitro [^89^Zr]-Atezolizumab Characterization

For specific binding studies (blocking study), 1 × 10^5^ MDA-MB-231 or HCC38 cells were seeded into a 12-well plate and allowed to adhere for 48 h. On the day of the experiment, [^89^Zr]-Atezolizumab was added to the cells (25 ng/100 μL; 3.7 kBq/100 μL) with (competitive binding) and without (total binding) presence of non-labeled atezolizumab (25 μg/100 μL) for 1 h. After incubation, cells were washed, trypsinized, and the radioactivity was assayed in a gamma counter (Wizard^2^, Perkin Elmer, Waltham, MA, USA). The percentage of [^89^Zr]-Atezolizumab bound in presence or absence of competitor was calculated and corrected for protein concentration.

The immunoreactivity of conjugates was evaluated in MDA-MB-231 cells using the Lindmo assay as previously described [27]. To evaluate the [^89^Zr]-Atezolizumab cellular internalization, 1 × 10^5^ MDA-MB-231 cells were seeded 48 h before the assay. On the day of the experiment, 1 mL of 25 ng/mL of [^89^Zr]-Atezolizumab was added and the plate was incubated for up to 24 h. At each time point, the wells were washed followed by the addition of 0.5 mL of cold 10 mM sodium citrate with 150 mM NaCl pH 3.0. After 5 min, the solution was carefully transferred into microcentrifuge tubes (surface bound fraction). The cells were trypsinized (internalized fraction), transferred into microcentrifuge tubes, and both fractions were counted in a gamma counter. The cell internalization ratio was calculated by fitting the results in a one-phase exponential curve [28].

### 2.4. In Vivo Assessment of [^89^Zr]-Atezolizumab

All animal experiments were performed with the approval of the Institutional Animal Care and Use Committee (IACUC) of the University of Alabama at Birmingham.

#### 2.4.1. Evaluation of In Vivo Specificity with PET Imaging and Biodistribution

Five-week-old female Balb/c NU/NU nude mice (Charles River; Wilmington, MA, USA) were subcutaneously implanted with 1 × 10^7^ MDA-MB-231 cells (100 µL) on the right shoulder and the tumors were allowed to grow for 3 weeks. MDA MB 231 tumor-bearing mice were injected, via tail vein, with 3.7 MBq (25 μg of antibody; non-competitive in vivo binding; N = 4) of [^89^Zr]-Atezolizumab or 3.7 MBq + 2.5 mg of unlabeled antibody (competitive in vivo binding; N = 4) and imaged 7 days post injection using a GNEXT small animal PET/CT (Sofie Biosciences, Culver City, CA, USA). Immediately after imaging, mice were euthanized; tumors and selected organs were collected, weighed, and analyzed in a gamma counter to measure radioactivity.

#### 2.4.2. Quantitative Analysis of [^89^Zr]-Atezolizumab Tumor Uptake as a Tool to Measure PD-L1 Expression

Five-week-old female NSG mice were purchased from Jackson Laboratories (Bar Harbor, ME, USA) and transplanted with TNBC PDX BCM 3936 [29] (Patient-derived Xenograft and Advanced In Vivo Models Core; Baylor College of Medicine; Houston, TX, USA) tumors into the 3rd mammary fat pad and the tumors were allowed to grow for 4 weeks. When the tumors were approximately 300 mm^3^ (298.9 mm^3^ ± 70.3 mm^3^), mice were enrolled in the experiment. BCM 3936 PDX tumor-bearing mice were injected retro-orbitally with 1.85 MBq [^89^Zr]-Atezolizumab (12 μg of radiolabeled antibody) and scanned 7 days post injection as previously described. Immediately following imaging, mice were euthanized, and tumors were excised and prepared for immunohistochemistry (method described in Appendix A) to evaluate PD-L1 expression as described above for biological analysis. On the imaging data, regions of interest (ROI) were identified for tumor and normal tissue (muscle) using CT anatomical reference, and Standard Uptake Values (SUV) of tissues of interest were calculated with VivoQuant image analysis software (version 3.5, Invicro, Boston, MA, USA). SUV_mean_ and histogram curves of SUV voxel distribution across tissues were calculated for each ROI. Voxels within the tumors more than 2 standard deviations above muscle with SUV_mean_ were considered positive for PD-L1 [30].

#### 2.4.3. Evaluation of [^89^Zr]-Atezolizumab-PET Imaging to Assess Changes in PD-L1 Expression Following Cytotoxic Treatment in TNBC Preclinical PDX Models

BCM 3936 PDX tumor-bearing mice were divided into 4 treatment groups (N = 4/group): niraparib (Zejula^®^; Medchemexpress; Monmouth Junction, NJ, USA), paclitaxel (Alfa Aesar; Haverhill, MA), radiation (X-RAD 320 irradiator; Precision X-ray, North Bradford, CT), or saline control. Five-week-old female NSG mice were transplanted with TNBC PDX BCM 3936 as previously described and allowed to grow to approximately 300 mm^3^ in size (308.9 mm^3^ ± 113.3 mm^3^). The niraparib group received 50 mg/kg of niraparib on day 0, 1, 2, and 3 via oral gavage; paclitaxel group received 10 mg/kg of paclitaxel on day 0 and 3 via IP injection; radiation group received 4 Gy of localized X-ray radiation (tumor) on day 0, 1, 2, and 3; and the control group received 100 μL of saline IP on day 0, 1, and 3. On day 7, mice were injected, intravenously, with 3.7 MBq of [^89^Zr]-Atezolizumab (24 μg of antibody) and scanned by PET/CT on days 0 (baseline), 1, and 4. Immediately following the last imaging point, tumors were collected and prepared for IHC and automatic PD-L1 quantification, as described in Appendix A.

##### Immunohistochemistry (IHC) and Quantification of PD-L1 Expression

Immediately following imaging, BCM 3936 PDX tumors were collected and fixed in 10% formalin followed by 70% ethanol for sectioning. Tumor sections were stained with a 1:100 dilution of anti-human PD L1 (ab210931, Abcam, Cambridge, MA, USA) overnight, followed by goat anti-mouse secondary antibody (VC002, R&D Systems, Minneapolis, MN, USA) for 3 h. Positive staining was visualized with HRP DAB substrate kit (SK-4100, Vector Laboratories, Burlingame, CA, USA) and nuclear counterstained with hematoxylin.

Custom MATLAB codes were developed to automatically quantify PD-L1+ IHC signaling and necrosis (H&E) in whole cross-sectional tumor slices. Whole tumor sections were imaged (EVOS M7000 Imaging, ThermoFisher; Waltham, MA, USA). Segmentation of diaminobenzideine and hematoxylin was accomplished by adapting the scikit-learn color deconvolution [31]. Afterward, a binary mask was formed by applying a locally adaptive threshold to each image. To minimize false-positive signals, nonconnected areas of less than 100 pixels were omitted. Positive IHC percentage was calculated as the fraction of the positive pixel count of total pixels within the segmented tumor section.

##### Dosimetry Estimation

Five-week-old female Balb/c mice (Charles River; Wilmington, MA, USA) were injected, via tail vein, with 2.59 MBq (100 μL) of [^89^Zr]-Atezolizumab and divided into 7 groups (N = 5). After 1 h, 1, 2, 6, 8, 12, and 14 days, one group was euthanized, and select organs were collected, weighed, and analyzed in a gamma counter. All data were normalized to the one-hour time point, and then the percent injected dose (%ID) was calculated. The area under the time vs. %ID curve was computed using a numerical integration method for each organ. Physical decay only was used from the final measured time point onward. The areas under the curves were entered in OLINDA/EXM Version 2.2 (Copyright 2012 Vanderbilt University, Nashville, TN, USA). The dose for each target organ as well as the effective dose was calculated in OLINDA using the ICRP 89 human adult female model provided in the OLINDA software. This method was adapted from the literature [9].

### 2.5. Statistical Analysis

Results were expressed as mean ± standard deviation (SD). For simple comparison between two groups, the Student’s *t*-test was used; for multiple comparisons, one-way ANOVA was performed followed by group comparisons using the Bonferroni method to control for multiple comparisons. Statistical analysis was performed using GraphPad Prism version 9 (GraphPad Software Inc., La Jolla, CA, USA) and *p* < 0.05 was considered statistically significant.

## 3. Results

### 3.1. Synthesis and In Vitro Characterization of [^89^Zr]-Atezolizumab

The optimal conjugation molar ratio (mAb:DFO) found was 1:20, which provided the highest radiochemical yield without additional purification (Figure 1A); therefore, this molar ratio was used for immunoreactivity studies. Chemistry optimization increased the radiochemical purity from 85.05% ± 0.74% (Figure 1A; 1:20; immediate) to 98.88% ± 1.3% (without additional purification) and showed stability >95% after 24 h at 2–8 °C (96.73% ± 0.49%) (Figure 1B). The specific activity of 0.15 MBq/µg was achieved with conservation of immunoreactivity (110.3% ± 29.8%).

The total binding of [^89^Zr]-Atezolizumab to MDA-MB-231 and HCC38 cells was 76.54% ± 2.23% and 30.50% ± 5.72%, respectively (Figure 1C). The uptake of [^89^Zr]-Atezolizumab was significantly reduced by the addition of non-labeled atezolizumab (blocking), in both cell types. Internalization assay revealed that 66.88% ± 1.03% of the total of [^89^Zr]-Atezolizumab added to the MDA-MB-231 cells was internalized after 24 h (Figure 1D).

### 3.2. Evaluation of Specificity In Vivo by PET/CT Imaging and Biodistribution in a Xenograft Mice Model

[^89^Zr]-Atezolizumab showed tumor uptake in MDA-MB-231 xenograft tumors with a specific binding, which was partially blocked by co-injection of non-radioactive antibody (2.5 mg of non-radioactive antibody; 25 µg of radioactive antibody; Figure 2A). Figure 2B shows representative images of mice undergoing blocking vs. non-blocking studies with decreased tumor uptake when blocked, which is consistent with biodistribution data. Significant increases in [^89^Zr]-Atezolizumab uptake are observed in the blood, heart, pancreas, skin, and tumor in mice that received a blocking dose prior to [^89^Zr]-Atezolizumab imaging (Figure 2C).

### 3.3. In Vivo Evaluation of the Potential of [^89^Zr]-Atezolizumab as a Tool to Measure PD-L1 Expression in TNBC PDX Mouse Models

To expand clinical relevance, [^89^Zr]-Atezolizumab was evaluated in a TNBC PDX model (BCM 3936). The tumor SUV_mean_ was significantly higher than control tissue (1.00 ± 0.16 SUV_mean_ vs. 0.39 ± 0.08 SUV_mean_; for tumor and muscle, respectively; *p* ≤ 0.0001; N = 5; Figure 3D), and PD-L1 expression in the tumor was confirmed by IHC (Figure 3C). The average fraction of PD-L1+ was 0.784 ± 0.171. Further, [^89^Zr]-Atezolizumab was shown to be tumor-specific when compared to muscle uptake, which revealed a 0.122 ± 0.132 average fraction of PD-L1 tissue (*p* = 0.0001; Figure 3A). Histogram analysis demonstrated that [^89^Zr]-Atezolizumab uptake is more heterogeneous within the tumor compared with control tissue as shown by the wider distribution of SUV values (Figure 3B and Appendix A).

### 3.4. Evaluation of [^89^Zr]-Atezolizumab as a Tool to Assess Changes in PD-L1 Expression Following Cytotoxic Treatment in TNBC PDX Mice Models In Vivo

Figure 4 shows the results obtained for the modulation study, where [^89^Zr]-Atezolizumab was injected into the TNBC PDX model, imaged at baseline, and treated with paclitaxel, niraparib, or radiation. A significant increase in tumor:muscle SUV_mean_ 1 and 4 days after therapy (following baseline imaging on day 0) with niraparib compared to the control group (2.069 ± 0.101 SUV_mean_ vs. 2.461 ± 0.320 SUV_mean_, for day 1 tumor and 2.098 ± 0.276 SUV_mean_ vs. 2.591 ± 0.224 SUV_mean_, for day 4; *p* ≤ 0.05; N = 5); however, paclitaxel and radiation only induced a slight upregulation (*p* > 0.05) of PD-L1 in the tumor. IHC Quantification of PD-L1 revealed that although there was no difference in PD-L1 expression within the groups (*p* > 0.05), all treatments induced target upregulation in the tumor (56.24% ± 12.33%; 59.02% ± 8.43%; 69.56% ± 10.75% and 66.53% ± 16.24% for control, paclitaxel, niraparib, and radiation groups, respectively). Niraparib induced upregulation compared to the control, which correlates with the results obtained with [^89^Zr]-Atezolizumab imaging.

### 3.5. Dosimetry Estimation

The activity curves for each organ obtained from the biodistribution studies were used for the extrapolation of effective doses in humans (Table 1).

The dosimetry study reveals the organs that will receive the higher dose (considering an adult human female model) are the spleen, adrenals, kidneys, and liver (3.78, 2.69, 2.54, and 2.49 mGy/MBq, respectively), with an effective dose of 0.773 mSv/MBq.

## 4. Discussion

Molecular imaging can help understand biological and molecular processes by providing whole-body, dynamic, and real-time expression of specific targets. Importantly, atezolizumab is FDA-approved for safety, providing a streamlined approach for future clinical use. While atezolizumab is no longer used as a therapeutic for TNBC, the need to quantify PD-L1 is still a clinically relevant question in TNBC. The ability to accurately screen for PD-L1, in a three-dimensional tumor, noninvasively, could provide a key tumor biological assessment to stratify patients that immune checkpoint inhibitors [11]. In this work, we describe the chemistry optimization and validation of [^89^Zr]-Atezolizumab as a molecular agent to identify PD-L1 expression in tumors, as well as its use to monitor changes in tumor PD-L1 expression induced by therapy. We are expanding upon a growing body of work that examines the potential of ^89^Zr-targeted PDL1 imaging probes. Jagoda, E. et al. and Bouleau, A. et al. evaluated [^89^Zr]-Avelumab and three different [^89^Zr]-labeled-Fab fragments in xenograft models of TNBC and lung cancer, respectively [19,32], and showed that ^89^Zr-Df-KN035 accumulates in the tumor and in key organs in a xenograft mice model [12]. While these studies showed promising results with tumor uptake and low non-target accumulation, the inclusion of a PDX model provides an opportunity to recreate the heterogeneity observed in clinical tumors. With this work, we aimed to add to the existing literature and evaluate the potential of [^89^Zr]-Atezolizumab in monitoring PD-L1 expression in a PDX mice model.

The chemistry optimization showed stability 24 h post labeling over 95%, conservation of immunoreactivity, and high in vitro uptake in cells with different expression levels of PD-L1 (Appendix A), which was consistent with the established differences in reported PD-L1 expression [33], and suggests that [^89^Zr]-Atezolizumab would be able to discern lesions with lower PD-L1 expression. Importantly, the long-term stability of similar radiolabeled antibodies has been previously demonstrated [23,26]; however, the longitudinal stability of [^89^Zr]-Atezolizumab should be assessed preclinically and clinically through iTLC. In vivo, [^89^Zr]-Atezolizumab demonstrated remarkable specificity, as demonstrated by its elevated uptake in tumors and organs that are acknowledged to express PD-L1, such as the spleen, lungs, heart, skin, and pancreas. The high spleen uptake was likely driven by the presence of macrophages, since PD-L1 was found to be highly expressed in macrophages of Balb/c nude mice [18,34], and B cells in this strain have been shown to be fully functional [35]. By co-injecting cold antibodies, the uptake in the spleen, blood, heart, and skin, increases, while the tumor uptake decreases (Figure 2A). One potential reason is that this occurs due to the saturation, by the blocking dose, of receptors in healthy tissues that express low levels of PD-L1, resulting in an increased quantity of radiotracer available in the bloodstream, as evidenced by its high blood retention. This surplus radiotracer is then more readily available to bind to organs expressing higher levels of PD-L1. Although the biodistribution blocking study evinced a reduction in tumor uptake and a marginal increase in spleen uptake, the imaging study showed the opposite trend to the biodistribution blocking study (Figure 2B). Specifically, we observed a slight, but not significant, increase in tumor SUV_mean_ (*p* = 0.3388) in the blocking group, concomitant with a decrease in spleen SUV_mean_ (*p* = 0.0437). While these preliminary findings warrant confirmation through replication in a larger cohort and validation via biodistribution studies, they align with previously reported results [17,36], including one conducted in non-human primates [37]. This observation suggests that PET imaging using [^89^Zr]-Atezolizumab can be performed in patients who are already receiving atezolizumab with no reduction in tumor uptake. Moreover, in a clinical study, Bensch et al. [13] co-injected 10 mg of cold atezolizumab into the patients and observed that the imaging signal corresponds to PD-L1 expression in various healthy lymphoid tissues and concluded that tracer uptake might be a strong predictor of response to atezolizumab treatment. This is consistent with other antibody-centric imaging probes that show intact antibodies require a cold loading dose to achieve optimal tumor visualization [38]. Bensch found a correlation between the non-malignant lymph node tissue and [^89^Zr]-Atezolizumab uptake.

In vitro, two TNBC cell lines (MDA MB 231 and HCC38) were used with a heterogeneous expression of PD-L1 [39,40]. Despite the difference in PD-L1 expression, cell lines with high PD-L1 expression (MDA MB 231) and low PD-L1 expression (HCC38) were used to highlight the sensitivity and specificity of our anti-PD L1 radiotracer. Since the MDA MB 231 cell line exhibits higher baseline expression of PD-L1, the internalization study and in vivo studies were conducted with this cell line. Since the use of PDX tumor models in preclinical studies can better recapitulate the tumor microenvironment and genetic heterogeneity observed in human cancers, we use a PDX model to evaluate the changes in PD-L1 expression by standard of care therapy. In this model, [^89^Zr]-Atezolizumab-PET imaging showed increases in PD-L1 expression induced by therapy, by presenting a significant increase in tumor:muscle SUV_mean_ 1 and 4 days after therapy with niraparib. Additionally, although we did not observe significant differences within the paclitaxel and radiation groups, there was a trend in the upregulation of PD-L1 in these groups. Jiao, S. et al. have shown that, by Western blot, flow cytometry, and IHC, PARP inhibitors upregulate PD-L1 expression in breast cancer cells and in a preclinical model of breast cancer [41]. As patients with TNBC are rarely treated with single therapeutic approaches alone (chemotherapy or radiation), studying how a combination of chemotherapy and radiation therapy affects the PD-L1 expression and, subsequently, how this combination sensitizes to immunotherapy would be of interest. Changes in [^89^Zr]-Atezolizumab could also be observed in the context of novel small molecule-based immunotherapies, such as PD1-VAXX or PD1-derived CA-170 therapy, to elucidate whether changes in PD-L1 could be predictive of the eventual response to these drugs [42,43,44]. This result suggests that [^89^Zr]-Atezolizumab may be used to identify the mechanisms of immunotherapy synergy. [^89^Zr]-Atezolizumab therapy studies have been conducted prior to and following paclitaxel, niraparib, or radiation therapy to study the dynamics of PD-L1 expression in TNBC tumors.

The dosimetry study indicates the endocrine tissues/organs receive the highest dose, which is expected since PD-L1 is physiologically expressed in these tissues. Liver and intestinal accumulation of the [^89^Zr]-Atezolizumab was consistent with primary clearance of antibodies [45]. The estimated human effective dose of 0.773 mSv/MBq, therefore, the administration of 37 MBq in patients, as already performed in a clinical study with [^89^Zr]-Atezolizumab [13], would result in total effective doses of 28.6 mSv. This dose is also similar to the values obtained in clinical PET studies using other ^89^Zr-labeled antibodies. For instance, Laforest, R. et al. (2016) reported an effective dose of 0.47 mSv/MBq in patients who received 62 MBq of [^89^Zr]Trastuzumab [46], Lindenberg, L. et al. (2017) reported an effective dose between 0.264 and 0.330 mSv/MBq for [^89^Zr]Panitumumab [47] and Ulaner, G. et al. (2018) estimated an effective dose of 0.54 mSv/MBq for [^89^Zr]Pertuzumab [48] Although most dosimetry studies for Zr-89 have been conducted in humans, there have been some preclinical studies in mice. For example, Kelly, M. et al. (2021) estimated an effective dose of 23.01 mSv in humans for [^89^Zr]DFO-REGN3504—an anti-PDL1 antibody—using animal models [9]. This favorable dosimetry may enable repeated PD-L1 PET imaging, as may be required for treatment response monitoring.

Limitations of the study include the absence of an extended-term evaluation of the radiotracer stability in human and mouse serum in vitro. Future works also should explore a long-term evaluation of tumor response to combination therapies in relation to PD-L1 expression. Although the present study showed promising results in identifying early response to the therapy, future studies would require a second administration of [^89^Zr]-Atezolizumab to study the late-therapy response (≥4 weeks after the first administration). Another limitation is that these studies were performed in athymic nude and NSG mice models and the translation of this technique will require testing in animals with an intact immune system.

## 5. Conclusions

[^89^Zr]-Atezolizumab showed potential for noninvasive detection of PD-L1-positive TNBC tumors as well as identifying tissue heterogeneity. Furthermore, this radiopharmaceutical showed promising results for its ability to monitor the dynamic expression of PD-L1 induced by PARP inhibitors and radiation therapy, which can help to identify the mechanisms of immunotherapy synergy. Lastly, this noninvasive imaging method might be useful in identifying patients who will respond to anti-PD-L1 therapy, overcoming IHC limitations.

## Figures and Tables

**Figure 1 cancers-15-02708-f001:**
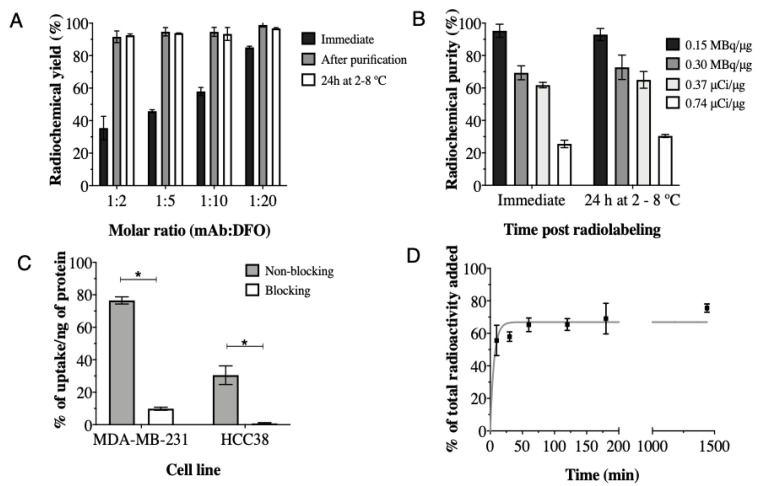
Immediate radiochemical yield and 24 h stability at 2–8 °C of atezolizumab conjugated to DFO at different molar ratios and radiolabeled with ^89^Zr at 0.15 MBq/μg, (**A**); Radiochemical purity of [^89^Zr]-Atezolizumab radiolabeled with ^89^Zr in different conditions (**B**); Cell uptake of [^89^Zr]-Atezolizumab in MDA-MB-231 and HCC38 cells (**C**) and internalization of [^89^Zr]-Atezolizumab in MDA-MB-231 cells in vitro (**D**). Results expressed in Mean ± SD; Means represent 3 or more independent experiments performed in triplicate. Significant differences between groups are indicated by * (*p* ≤ 0.05).

**Figure 2 cancers-15-02708-f002:**
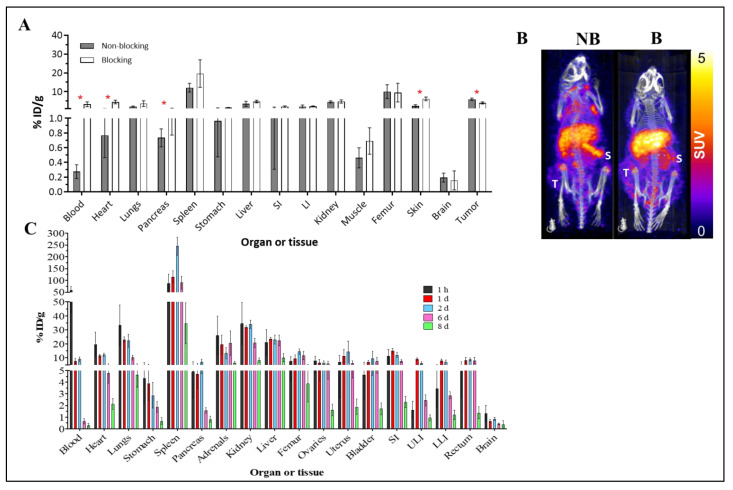
Biodistribution of [^89^Zr]-Atezolizumab in Balb/c nude mice bearing MDA-MB-231 tumors. Blocking group received 2.5 mg of unlabeled atezolizumab concurrently with [^89^Zr]-Atezolizumab and were euthanized 7 days p.i. (N = 4/group) (**A**); Representative PET-CT imaging of [^89^Zr]-Atezolizumab in mice bearing MDA-MB-231 tumors during a blocking experiment (B = blocking, NB = non-blocking) at 7 day p.i. revealing marginal increased signal accumulation in the spleen (S) during the blocking study (N = 2/group); Tumor uptake is denoted with (T) (**B**); Biodistribution of [^89^Zr]-Atezolizumab in non-bearing tumor Balb/c mice for dosimetry estimation from 1 h to 8 days p.i. (N = 5/group) (**C**). Results expressed in Mean ± SD. SI = Small Intestines; ULI = Upper large intestines; LLI = Lower large intestines; * *p* ≤ 0.05.

**Figure 3 cancers-15-02708-f003:**
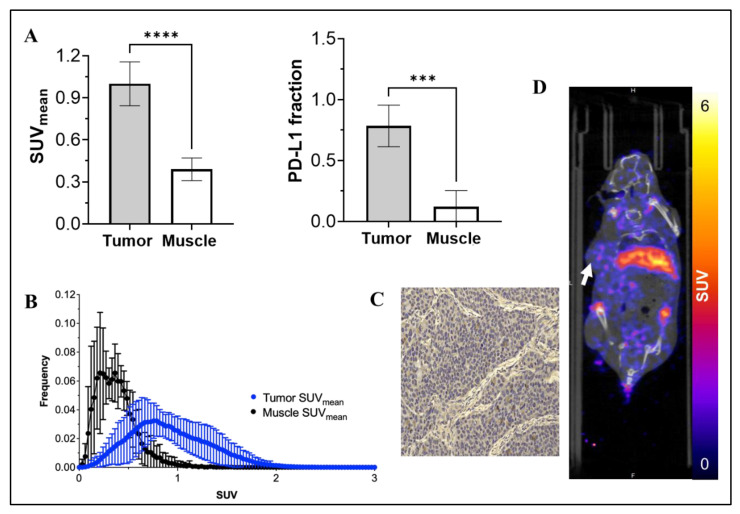
SUV_mean_ of [^89^Zr]-Atezolizumab uptake in tumor and muscle, and a fraction of PD-L1 expressing cells in tumor and muscle (**A**); Voxel histogram analysis of tumor and muscle (**B**); Representative IHC slide of tumor stained with anti-PD-L1 antibody (**C**). More IHC images in larger size are available in the Appendix A; and Representative PET-CT imaging of [^89^Zr]-Atezolizumab in NSG mice transplanted with TNBC PDX BCM 3936 (**D**). White arrow denotes the location of the tumor xenograft. Results expressed in Mean ± SD; N = 5; *** *p* = 0.001; **** *p* < 0.0001.

**Figure 4 cancers-15-02708-f004:**
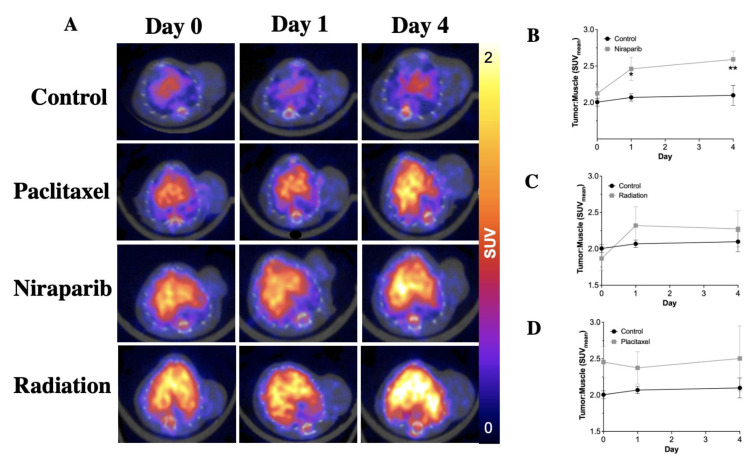
Representative cross section of [^89^Zr]-Atezolizumab PET images of TNBC PDX mice model showing update in the tumor (white arrows) and liver (black L) after treatment with paclitaxel, niraparib, and radiation (**A**); Measurement of PD-L1 expression in tumor of TNBC PDX mice model after treatment with paclitaxel (**B**), niraparib (**C**), and radiation (**D**). Results demonstrate significant differences in PD-L1 uptake in Niraparib-treated tumors, as well as trending differences in PD-L1 in Paclitaxel or radiation-treated groups. Results expressed in Mean ± SD; N = 4/group. * *p* < 0.05, ** *p* < 0.01.

**Table 1 cancers-15-02708-t001:** Estimated absorbed dose in target organs calculated through the extrapolation of mouse data to human, by using the ICRP 89 human adult female model provided in the OLINDA software.

Target Organ	Absorbed Dose (mGy/MBq)
Spleen	3.78
Adrenals	2.69
Kidneys	2.54
Liver	2.49
Ovaries	2.02
Small Intestine	1.82
Left colon	1.18
Pancreas	1.17
Gallbladder Wall	1.08
Right colon	1.06
Uterus	1.02
Stomach Wall	0.98
Heart Wall	0.64
Esophagus	0.59
Lungs	0.57
Rectum	0.52
Red Marrow	0.48
Thymus	0.45
Osteogenic Cells	0.38
Urinary Bladder Wall	0.37
Breasts	0.32
Thyroid	0.32
Salivary Glands	0.26
Eyes	0.23
Brain	0.11

## Data Availability

Datasets and materials are available upon reasonable request.

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
