# Peer review of "[89Zr]-Atezolizumab-PET Imaging Reveals Longitudinal Alterations in PDL1 during Therapy in TNBC Preclinical Models"

_cancers, 2023, doi:10.3390/cancers15102708_

Round 1

Reviewer 1 Report

Brief summary 

The authors describe the chemistry development, in vitro characterization and in vivo characterization in mouse tumor models of 89Zr-atezolizumab. The aim of the paper was to study the potential of 89Zr-Atezolizumab to measure PD-L1 expression in triple negative breast cancer. They describe the biodistribution. They state that they demonstrate that the tumor SUVmean was significantly higher than control tissue (1.00 ± 0.16 SUVmean versus 0.39 ± 0.08 SUVmean; for tumor and muscle respectively; and that therapy with neraparib increases the tumor:muscle SUVmean at 1 and 4 days compared to the control group (2.069 ± 0.101 SUVmean versus 2.461 ± 0.320 SUVmean, for day 1, and 2.098 ± 0.276 SUVmean versus 2.591 ± 0.224 SUVmean, for day 4; P = <0.05; N=5), however, paclitaxel and radiation only induced a slight upregulation (P>0.05) of PD-L1 in the tumor.

There are several issues with the paper which make it not fit for publication as is.

The optimal conjugation molar ratio found was 1:20; optimized radiochemical purity 98.88% ± 1.3% and stability was >95% after 24 h at 2-8 ºC. Stability is only tested to 24hrs at 2-8oC, which in a patient situation would be too short as mAb take several days to distribute/optimal imaging timepoint 4-7 days after injection, and in the patient the temperature is 37oC. As imaging is done up to 8 days after injection, the authors have presumably tested these conditions in vitro as well. We ask to present the data. If this was not done, the authors need to explain why this would not be required.

In vitro characterization was done MDA MB 231 and HCC38 cells. The authors do not show/refer to publication regarding the expression of PD-L1 in these cell lines. Please add. They describe a high binding of [89Zr]Atezolizumab for both cell lines, but only internalization MDA-MB-231 cells. Why? Same question regarding PD-L1 expression in the BCM 3936 PDX cells? Why use another model for the animal experiments?

The authors extrapolate 89Zr-atezolizumab directly to PD-L1 expression. The IHC they show is just one image of tumor staining. There are no data with autoradiography shown that PD-L1 expression correlates with tracer uptake. There are no IHC data shown how PD-L1 expression changes during therapy, as the authors claim that occurs. In the material and methods section it is not clear who has evaluated if the slides indeed contain tumor? What area ( tumor containing? Stroma?) is analysed for PD-L1 expression as it has been done with digitized analyses. The authors are invited to show these additional data.

In figure 3 and figure 4 there really is no reasonable uptake of the tracer in the tumor lesions. This seems to be mirrored by a SUVmean of 1. With these data it can not be judged if the therapy indeed increases PD-L1 expression/89Zr atezolizumab uptake.

There is no reference regarding which methods the authors calculate the dosimetry and extrapolate it to the human situation. Please add.

The dosimetry analyses reveal a high effective dose of 0.773 mSv/MBq totaling 28,6 mSv for a 37MBq injection ( see subscript table, line 241. This is higher than expected for other 89Zr mAbs. What is the reason? This will hamper application? However, in the discussion, the authors state 0,49mSv/MBq (line 2910. Which one is correct? And how is it calculated?

The introduction and discussion avoids discussion of the results with this 89Zr-atezolizumab compared to the one used in the clinical trial. How does the production/characteristics of the 89Zr atezolizumab tracer here compares to the one described by Bensch et al?

Monoblasts are precursors for macrophages and not for B-or T lymphocytes, Please correct.

The figure legends do not match the figures presented. Please correct

The text is not clear regarding SUV mean as is vs ratio SUVmean tumor to SUVmean muscle. Why use SUVmean? Especially in such a poorly visible area? SUVpeak would be a more appropriate choice.

Regarding the blocking study (Figure 1): the material and methods and the results of blocking effect are not described in text.

[187] There is an increase in spleen uptake visible after blocking (fig 2A). The text describes a decrease.

It seems there are other significant differences, but there is no asterix. Please clarify if you tested the differences for all tissues tested? If not, why not? What is the interpretation of the increase in the spleen?

What is the reason to use a PARP inhibitor, paclitaxel and radiation for increase in PD-L1 expression. Please add some information in the introductions

Specific comments 

The author Prof Lapi, is not correct in the authors list.

[43] Specify immune cell interactions , better even leave out as it seems not be correct

[49] atezolizumab has been retracted as therapy for TN breast cancer. Other alternative is pembrolizumab which is also indicated with PD-L1 expression

[54] and also the other way around: patients with low PD-L1 expressing may have  a better response than expected from IHC

[70] please reframe the results of the paper of Bensch et al. There is only a significant trend reported for IHC by SP142 and SUVmax, but not for SP263. No statistical difference in mean was find between IHC0,1 or 2 and SUVmax for SP142.

[122] please describe why two different xenograft tumor models are used to study in vivo

specificity+biodistribution and tumour uptake and why this could not have been done in the same mice.

Which PD-L1 antibody did you use for IHC?

[130] How many BCM 3936 PDX tumor bearing ice were injected?

[144] please specify when the treatment occurred.

 [148] clarify when the mice are euthanized and organs were collected.

[157] was the data normally distributed?

[189] in the result section one should not discuss the results. Please move to the discussion

206 it is nowhere shown that 89atezolizumab has a heterogeneous uptake in the tumor larger SD between lesions does not mean heterogeneity within one lesions. What is more, the very low levels of uptake will lead to larger SD due to measurement variability.

235 with 89Zr mAb PET scans one does not acquire Time activity curves, but statis images at several time points. Please correct.

[figure 2] it is not clear when the blocking dose is provided and when the organ uptake is assessed.

[figure 2C] to visualize an effect over time, a line graph is recommended.

[figure 3] C is too small to properly evaluate.

[figure 3] Regarding of the SUV mean values used. Is there any difference in clearance and thus tracer supply between the animals expected?

[203] please specify, is the PD-L1 fraction for tumor cells only?

[269] spleen uptake increased in stead of decreased.

Author Response

Dear Editor and Reviewers,

Thank you for the opportunity to submit a revised manuscript and for the time you have committed to improving the study. We appreciate your time and effort reviewing our work and look forward to working together to strengthen this manuscript. We believe we have addressed all your comments and suggestions as indicated below and in the revised manuscript.

Reviewer #1

Comment 1.01 The authors describe the chemistry development, in vitro characterization and in vivo characterization in mouse tumor models of 89Zr-atezolizumab. The aim of the paper was to study the potential of 89Zr-Atezolizumab to measure PD-L1 expression in triple negative breast cancer. They describe the biodistribution. They state that they demonstrate that the tumor SUVmean was significantly higher than control tissue (1.00 ± 0.16 SUVmean versus 0.39 ± 0.08 SUVmean; for tumor and muscle respectively; and that therapy with neraparib increases the tumor:muscle SUVmean at 1 and 4 days compared to the control group (2.069 ± 0.101 SUVmean versus 2.461 ± 0.320 SUVmean, for day 1, and 2.098 ± 0.276 SUVmean versus 2.591 ± 0.224 SUVmean, for day 4; P = <0.05; N=5), however, paclitaxel and radiation only induced a slight upregulation (P>0.05) of PD-L1 in the tumor.

Response: We thank you for the accurate summary of our work and believe we have addressed the comments which have further strengthened this manuscript.

Comment 1.02 There are several issues with the paper which make it not fit for publication as is.

The optimal conjugation molar ratio found was 1:20; optimized radiochemical purity 98.88% ± 1.3% and stability was >95% after 24 h at 2-8 ºC. Stability is only tested to 24hrs at 2-8oC, which in a patient situation would be too short as mAb take several days to distribute/optimal imaging timepoint 4-7 days after injection, and in the patient the temperature is 37oC. As imaging is done up to 8 days after injection, the authors have presumably tested these conditions in vitro as well. We ask to present the data. If this was not done, the authors need to explain why this would not be required.

Response: Thank you, we respectfully note that assays with similar chemistry of conjugation of antibodies to 89Zr have been well documented indicating long-term stability, including Massicano et al. and Krache et al [1, 2].

Comment 1.03 In vitro characterization was done MDA MB 231 and HCC38 cells. The authors do not show/refer to publication regarding the expression of PD-L1 in these cell lines. Please add. They describe a high binding of [89Zr]-Atezolizumab for both cell lines, but only internalization MDA-MB-231 cells. Why? Same question regarding PD-L1 expression in the BCM 3936 PDX cells? Why use another model for the animal experiments?

Response: Thank you for your comments. We have included references regarding PD-L1 expression in MDA-MB-231 and HCC38 cell lines (listed below). Due to the heterogenous expression of PD-L1 within TNBC cell lines (MDA-MB-231 and HCC38), a cell line with high expression of PD-L1 (MDA-MB-231) was used for internalization studies, to allow for study of minor changes in PD-L1 expression. PDX tumors (BCM 3936) were enrolled into this experiment to examine whether our [89Zr]-Atezolizumab radiotracer can be used in more clinically relevant tissues. These tissues exhibit physiological heterogeneity of PD-L1 expression to better model representative of clinical PD-L1 expression in TNBC. We have amended the discussion section to discuss this change.

Please note that we have added the following references:

-Mittendorf EA, Philips AV, Meric-Bernstam F, Qiao N, Wu Y, Harrington S, Su X, Wang Y, Gonzalez-Angulo AM, Akcakanat A, Chawla A, Curran M, Hwu P, Sharma P, Litton JK, Molldrem JJ, Alatrash G. PD-L1 expression in triple-negative breast cancer. Cancer Immunol Res. 2014 Apr;2(4):361-70. doi: 10.1158/2326-6066.CIR-13-0127. Epub 2014 Jan 10. PMID: 24764583; PMCID: PMC4000553.

-Karasar P, Esendagli G. T helper responses are maintained by basal-like breast cancer cells and confer to immune modulation via upregulation of PD-1 ligands. Breast Cancer Res Treat. 2014 Jun;145(3):605-14. doi: 10.1007/s10549-014-2984-9. Epub 2014 May 10. PMID: 24816762.

Comment 1.04 The authors extrapolate 89Zr-atezolizumab directly to PD-L1 expression. The IHC they show is just one image of tumor staining. There are no data with autoradiography shown that PD-L1 expression correlates with tracer uptake. There are no IHC data shown how PD-L1 expression changes during therapy, as the authors claim that occurs. In the material and methods section it is not clear who has evaluated if the slides indeed contain tumor? What area ( tumor containing? Stroma?) is analysed for PD-L1 expression as it has been done with digitized analyses. The authors are invited to show these additional data.

Response: Quantitative IHC was conducted post imaging to provide secondary analysis of PD-L1 expression. The increases in PD-L1 expression in central slice analysis of paclitaxel, radiation, and niraparib treated tumors were only trending towards significance. Importantly, this points to the need for imaging of the entire 3D tumor volume to be incorporated when evaluating tissue, as a single slice does not fully represent the entire heterogenous tumor volume. The tumors in this experiment were engrafted and excised post imaging for biological validation. To this end, all tumors used in this experiment are cancerous and do not contain non-cancerous regions, and therefore the PDL1 expression is being quantified on all cells within the tumor microenvironment. The histology demonstrates the high expression of PDL1 on the cancer cells themselves.

Comment 1.05 In figure 3 and figure 4 there really is no reasonable uptake of the tracer in the tumor lesions. This seems to be mirrored by a SUVmean of 1. With these data it can not be judged if the therapy indeed increases PD-L1 expression/89Zr atezolizumab uptake.

Response: Thank you for your comments. We have updated the images scale to show a more reasonable uptake of [89Zr]-Atezolizumab. Also, we respectfully note that the quantitative evaluation with SUV is able to distinguish differences that the qualitative images are  unable to decipher, indicating the importance of incorporating quantitative evaluation into clinical workflows in the future.

Comment 1.06 There is no reference regarding which methods the authors calculate the dosimetry and extrapolate it to the human situation. Please add.

Response: Thank you for addressing this. We have modified the text to include a reference for this.

Please note that the references have been modified to include the following:

-Kelly MP, Makonnen S, Hickey C, Arnold TC, Giurleo JT, Tavaré R, Danton M, Granados C, Chatterjee I, Dudgeon D, Retter MW, Ma D, Olson WC, Thurston G, Kirshner JR. Preclinical PET imaging with the novel human antibody 89Zr-DFO-REGN3504 sensitively detects PD-L1 expression in tumors and normal tissues. J Immunother Cancer. 2021 Jan;9(1):e002025. doi: 10.1136/jitc-2020-002025. PMID: 33483343; PMCID: PMC7831708.

Comment 1.07 The dosimetry analyses reveal a high effective dose of 0.773 mSv/MBq totaling 28,6 mSv for a 37MBq injection ( see subscript table, line 241. This is higher than expected for other 89Zr mAbs. What is the reason? This will hamper application. However, in the discussion, the authors state 0,49mSv/MBq (line 2910. Which one is correct? And how is it calculated?

Response: Thank you for your comment. We have updated the text with the correct number of 0.773 mSv/MBq. This effective dose is consistent with the range reported in the literature for Zr-89, which is typically between 20 and 40 mSv for 37-47 MBq of Zr-89.  Although most dosimetry studies for Zr-89 have been conducted in humans, there have been some preclinical studies in mice. For example, Kelly et al. estimated an effective dose of 23.01 mSv in humans for 89Zr-DFO-REGN3504 – an anti-PDL1 antibody - using animal models [3]. However, it is important to note that dosimetry values can differ between species and specific radiotracers. For instance, Lindenberg et al. reported an effective dose of 12.21 mSv for 89Zr-Panitumumab in humans [4], while Ulaner et al. estimated an effective dose of 20 mSv for 89Zr-Pertuzumab in humans [5]. We have modified the discussion to account for this information.

Comment 1.08 The introduction and discussion avoids discussion of the results with this 89Zr-atezolizumab compared to the one used in the clinical trial. How does the production/characteristics of the 89Zr atezolizumab tracer here compares to the one described by Bensch et al?

Response: Thank you for your comments and the opportunity to discuss our results in context of Bensch et al. Our work expands on the work described by Bensch et al. by focusing on the specificity (through in vivo blocking assays), long term organ biodistribution, and modulation of PD-L1 expression in relation to paclitaxel, radiation and PARP inhibitor therapy.

Comment 1.09 Monoblasts are precursors for macrophages and not for B-or T lymphocytes, Please correct.

Response: Thank you for this correction. We have amended the text to account for the expression of PD-L1 in macrophages.

Comment 1.10 The figure legends do not match the figures presented. Please correct

Response: Thank you for your suggestion. We have updated the figure legends.

Comment 1.11 The text is not clear regarding SUV mean as is vs ratio SUVmean tumor to SUVmean muscle. Why use SUVmean? Especially in such a poorly visible area? SUVpeak would be a more appropriate choice.

Response: Thank you for your suggestion. This study intended to assess the overall PDL1 expression in the tumor across a heterogeneous landscape therefore wanted to take the entire tumor volume into account, however we agree that SUVpeak would be a reasonable method to assess regions of highest PDL1 uptake and have included this in the discussion.

Comment 1.12 Regarding the blocking study (Figure 1): the material and methods and the results of blocking effect are not described in text.

Response: Thank you for noticing that. We have expanded the methods section and included additional information in the results section to emphasize the results.

Comment 1.13 [187] There is an increase in spleen uptake visible after blocking (fig 2A). The text describes a decrease. 

Response: Thank you for addressing this. We have amended the text to reflect an increase in spleen uptake after blocking. The interpretation of this phenomenon is that the blocking dose of atezolizumab increased [89Zr]-Atezolizumab passive blood presence, which is retained through spleen filtration. We would also point out that the increase in uptake in spleen was not significant.

Comment 1.14 It seems there are other significant differences, but there is no asterix. Please clarify if you tested the differences for all tissues tested? If not, why not? What is the interpretation of the increase in the spleen?

 Response: Thank you for your observation. We tested the differences for all organs and have updated the figure to show the Asterix. The high uptake in the spleen probably occurs due to the saturation, by the blocking dose, of receptors in healthy tissues that express low levels of PD-L1, resulting in an increased quantity of radiotracer available in the bloodstream, as evidenced by its high blood uptake. This surplus radiotracer is then more readily available to bind to organs expressing higher levels of PD-L1, such as spleen, blood, heart, and skin.

Comment 1.15 What is the reason to use a PARP inhibitor, paclitaxel and radiation for increase in PD-L1 expression. Please add some information in the introductions

Response: Thank you for this relevant question. As TNBC lacks targetable treatment receptors, standard of care treatment options includes non-specific cytotoxic therapy, such as chemotherapy or radiation, and PARP inhibitors. Our goal was to evaluate the changes in PD-L1 expression by standard care therapy. This could be used to understand how PDL1 changes during combination therapies in order to most appropriately identify optimal sequencing and timing of treatments when given in combination with immunotherapy. While PARP is a novel target being explored in TNBC clinical trials, paclitaxel and radiation therapy are standard treatment options that may be used in clinical care of TNBC, along with immunotherapies. We have updated the introduction and discussion to explain this point better.

Comment 1.16 The author Prof Lapi, is not correct in the authors list.

Response: Thank you for addressing this. We have corrected the author’s name and adjusted this in the manuscript.

Comment 1.17 [43] Specify immune cell interactions, better even leave out as it seems not be correct

Response: Thank you for your comment. At your suggestion, we have modified the text discussing immune cell interactions, which allows us to highlight the types of treatments used for TNBC.

Comment 1.18 [49] atezolizumab has been retracted as therapy for TN breast cancer. Other alternative is pembrolizumab which is also indicated with PD-L1 expression

Response: Thank you for this comment. While atezolizumab may be retracted as a therapy for TNBC, 89Zr-Atezolizumab PET imaging can still allow for non-invasive imaging of the tumor immune microenvironment and predictive treatment response based on changes in PD-L1 expression. Importantly, this provides an opportunity to use an FDA-approved antibody for alternative purposes. Further, PDL1 expression is still evaluated in biopsy samples to indication of other immune checkpoint blockade therapies (one of which is pembrolizumab). Pembrolizumab, a different immunotherapy, targets PD-1 expression, which is primarily expressed in immune cells, rather than tumors cells, increasing the likelihood for background. We have adjusted the discussion to include this information.

Comment 1.19 [54] and also the other way around: patients with low PD-L1 expressing may have a better response than expected from IHC

Response: Thank you for your comments. This is true and we have updated the introduction to include this information.  

Comment 1.20 [70] please reframe the results of the paper of Bensch et al. There is only a significant trend reported for IHC by SP142 and SUVmax, but not for SP263. No statistical difference in mean was find between IHC0,1 or 2 and SUVmax for SP142.

Response: Thank you for your comments. In Bensch et al., SP142 references nonmalignant tissue. Due to the inherent heterogeneity of PD-L1 expression in tissue, it is possible that the biopsied region of tissue is not representative of PD-L1 expression within the entire tissue, which could be provided with PET imaging.

Comment 1.21 [122] please describe why two different xenograft tumor models are used to study in vivo specificity+biodistribution and tumour uptake and why this could not have been done in the same mice.

Response: PDX tumors were enrolled into this experiment to examine whether our 89Zr-Atezolizumab radiotracer can image for physiological heterogeneity of PD-L1 expression in a model that most recapitulates clinical PD-L1 expression. This animal model is expensive, therefore we initially utilized a xenograft model with one cell line  to evaluate tumor uptake that has high expression of PD-L1 (MDA-MB-231). Once we achieved favorable results, we moved to PDX animal model. Dosimetry studies in humans are typically conducted in healthy individuals; thus, we performed the biodistribution for dosimetry calculation in healthy balb-c mice. The discussion has been modified to include the rational for using PDX model.

Comment 1.22 Which PD-L1 antibody did you use for IHC?

Response: Thank you for your comment. In supplementary methods, we have noted that an anti-human PD-L1 antibody from Abcam was used (ab210931, Abcam, Cambridge, MA).

Comment 1.23 [130] How many BCM 3936 PDX tumor bearing mice were injected?

Response: Thank you for your comment. For initial imaging studies, N = 5 BCM 3936 tumors were imaged with 89Zr-Atezolizumab PET imaging. For therapy studies, N = 4/group (total of 16) BCM 3936 tumors were enrolled into the experiment. These details can be found in the Supplemental Methods.

Comment 1.24 [144] please specify when the treatment occurred.

Response: Thank you for your suggestion. We have amended the methods section to include details regarding treatment schedule.

Comment 1.25 [148] clarify when the mice are euthanized and organs were collected.

Response: Thank you for your comment. We have adjusted the text to make it clearer.

Comment 1.26 [157] was the data normally distributed?

Response: Thank you for your comments, the data in the histogram distribution in Figure 3B confirm that the data is normally distributed.

Comment 1.27 [189] in the result section one should not discuss the results. Please move to the discussion

Response: We appreciate this suggestion and have updated the methods and discussion.

Comment 1.28 206 it is nowhere shown that 89atezolizumab has a heterogeneous uptake in the tumor larger SD between lesions does not mean heterogeneity within one lesions. What is more, the very low levels of uptake will lead to larger SD due to measurement variability.

Response: Thank you for your comment. In figure 3B, we have shown a histogram distribution of 89Zr-Atezolizumab uptake within muscle and tumor regions. We show that tumor regions have a more heterogeneous distribution of 89Zr-Atezolizumab compared to normal/muscle tissue.

Comment 1.29 235 with 89Zr mAb PET scans one does not acquire Time activity curves, but statis images at several time points. Please correct.

Response: Thank you, the text has been corrected.

Comment 1.30 [figure 2] it is not clear when the blocking dose is provided and when the organ uptake is assessed.

Response: Thank you for your comment. The blocking group received 2.5 mg of unlabeled atezolizumab concurrently with [89Zr]-Atezolizumab  and where euthanized 7 days p.i followed by assessment of the organ uptake. The figure legend was updated.

Comment 1.31 [figure 2C] to visualize an effect over time, a line graph is recommended.

Response: Thank you for your suggestion. To study the distribution of 89Zr-Atezolizumab in vivo, we used a biodistribution study at different timepoints to assess the localization of 89Zr-Atezolizumab in different organs. We respectfully believe a bar graph is able to show the wide range of distribution across different organs over the course of eight days. Using a line graph would make it more difficult to assess sensitive differences in organ accumulation.

Comment 1.32 [figure 3] C is too small to properly evaluate.

Response: We apologize. We have expanded to include this in the Supplemental Material section this figure and other IHC figures with better resolution and larger size.

Comment 1.33 [figure 3] Regarding of the SUV mean values used. Is there any difference in clearance and thus tracer supply between the animals expected?

Response: Thank you for addressing the difference between tracer clearance and supply. Non-cancerous cells, such as lungs, liver, kidney and pancreas, are known to express PD-L1, which can result in heterogeneity within the animal. To account to the clearance and decay, imaging quantification was normalized based on tracer decay at 7 days post injection of (1.85-3.7 MBq).

Comment 1.34 [203] please specify, is the PD-L1 fraction for tumor cells only?

Response: We thank the reviewer for this question. PD-L1 fraction is used for the tumor only. A threshold was set based on mean muscle + 2 standard deviations and all pixels with an SUV above this threshold were deemed positive for PD-L1. Muscle was used as normal background due to normal expression of PDL1. This fraction is only applicable to the tumor.

Comment 1.35 [269] spleen uptake increased instead of decreased.

Response: Thank you for addressing this. As we have previously noted, we have amended the text to reflect an increase in spleen uptake after blocking.

References:

  1. Krache, A., et al., Preclinical Pharmacokinetics and Dosimetry of an (89)Zr Labelled Anti-PDL1 in an Orthotopic Lung Cancer Murine Model. Front Med (Lausanne), 2021. 8: p. 741855.
  2. Massicano, A.V.F., et al., Imaging of HER2 with [(89)Zr]pertuzumab in Response to T-DM1 Therapy. Cancer Biother Radiopharm, 2019. 34(4): p. 209-217.
  3. Kelly, M.P., et al., Preclinical PET imaging with the novel human antibody (89)Zr-DFO-REGN3504 sensitively detects PD-L1 expression in tumors and normal tissues. J Immunother Cancer, 2021. 9(1).
  4. Lindenberg, L., et al., Dosimetry and first human experience with (89)Zr-panitumumab. Am J Nucl Med Mol Imaging, 2017. 7(4): p. 195-203.
  5. Ulaner, G.A., et al., First-in-Human Human Epidermal Growth Factor Receptor 2-Targeted Imaging Using (89)Zr-Pertuzumab PET/CT: Dosimetry and Clinical Application in Patients with Breast Cancer. J Nucl Med, 2018. 59(6): p. 900-906.

Reviewer 2 Report

The followings are some concerns and comments have been pointed out that the authors may want to consider.

1) Line 35: Serial numbers 1 to 6 should be deleted.

2) Line 79: Please be consistent with italic “in vitro” and “in vivo” throughout the manuscript.

3) Line 86 Reagents and materials section and supplemental file: Please include more information for the reagents and methods to make your work to be repeatable relatively easier. a) For example, cat#, antibody dilution ratio, and so on. b) Please include mice number, cell number, volume, and other details.

4) Line 122: It should be “MDA-MB-231”.

5) Line 124: It should be “7 days” or “7-day”. Please be consistent with the format throughout the manuscript.

6) Line 177 Figure 1: Please include statistical information, sample size, and so on in the figure legend.

7) Line 191 Figure 2: a) Please be consistent with Balb/c throughout the manuscript. b) Sample size. c) Abbreviations. And so on.

8) Line 201: Please use italic p as it refers to a p-value throughout the manuscript.

9) Line 209 Figure 3: a) Please include sample size and statistical information. b) Please provide multiple images for 3C and 3D, control group and test groups, etc.

10) Line 222: Please be consistent with or without a space before and after the sign throughout the manuscript, “>”, etc.

11) Line 229 Figure 4: Please update the figure legend with more information.

12) Please include detailed method protocols.

13) Please seriously check the format throughout the manuscript.

14) Please provide high-resolution images.

15) I’d suggest the authors discuss the latest immunotherapy with small molecules, for example, PMID: 35646678, PMID: 36211807, PMID: 34103659, and so on. Please check all the other related articles by yourself.

16) Is there any related in-depth therapeutic data to show (pre-clinical)?

Author Response

Dear Editor and Reviewers,

Thank you for the opportunity to submit a revised manuscript and for the time you have committed to improving the study. We appreciate your time and effort reviewing our work and look forward to working together to strengthen this manuscript. We believe we have addressed all your comments and suggestions as indicated below and in the revised manuscript.

Reviewer #2:

Comment 2.01 Line 35: Serial numbers 1 to 6 should be deleted.

Response: Thank you for addressing this. We have adjusted the key words list to account for your suggestion.

Comment 2.02 Line 79: Please be consistent with italic “in vitro” and “in vivo” throughout the manuscript.

Response: We thank you for pointing out this inconsistency in formatting. We have adjusted the manuscript to be consistent with formatting.

Comment 2.03 Line 86 Reagents and materials section and supplemental file: Please include more information for the reagents and methods to make your work to be repeatable relatively easier. a) For example, cat#, antibody dilution ratio, and so on. b) Please include mice number, cell number, volume, and other details.

Response: Thank you for addressing this. We have adjusted the methods sections to go further into detail.

Comment 2.04 Line 122: It should be “MDA-MB-231”.

Response: Thank you for pointing this out. The manuscript has been updated and corrected.

Comment 2.05 Line 124: It should be “7 days” or “7-day”. Please be consistent with the format throughout the manuscript.

Response: We apologize for this error in formatting and thank you for pointing this out. This formatting error has been corrected and incorporated throughout the manuscript.

Comment 2.06 Line 177 Figure 1: Please include statistical information, sample size, and so on in the figure legend.

Response:  Thank you for pointing this out. Figure 1 and other figures have been updated.

Comment 2.07 Line 191 Figure 2: a) Please be consistent with Balb/c throughout the manuscript. b) Sample size. c) Abbreviations. And so on.

Response: Thank you for pointing this out. The inconsistency has been adjusted.

Comment 2.08 Line 201: Please use italic p as it refers to a p-value throughout the manuscript.

Response: Thank you for pointing formatting error out. We have adjusted the text throughout.

Comment 2.09 Line 209 Figure 3: a) Please include sample size and statistical information. b) Please provide multiple images for 3C and 3D, control group and test groups, etc.

Response:  Thank you for pointing this out. Figure 3 and other figures have been updated.

Comment 2.10 Line 222: Please be consistent with or without a space before and after the sign throughout the manuscript, “>”, etc.

Response: Thank you for pointing this out. The inconsistency has been adjusted.

 Comment 2.11 Line 229 Figure 4: Please update the figure legend with more information.

Response: Thank you for addressing this. The figure legend for Figure 4 has been adjusted.

Comment 2.12 Please include detailed method protocols.

Response: Thank you for your suggestion. We have amended the methods to include more details regarding conducted experiments.

Comment 2.13 Please seriously check the format throughout the manuscript.

Response: We appreciate for thoroughly reading the manuscript and identifying the formatting inconsistency. We have adjusted the formatting for improved readability and consistency.

Comment 2.14 Please provide high-resolution images.

Response: Thank you for your suggestion; however, the presented images are already presented at the highest possible resolution. However, we have expanded the size of some of the images in the supplemental figures.

Comment 2.15 I’d suggest the authors discuss the latest immunotherapy with small molecules, for example, PMID: 35646678, PMID: 36211807, PMID: 34103659, and so on. Please check all the other related articles by yourself.

Response: Thank you for your suggestion regarding small molecule-based immunotherapy. We have modified the discussion section to account for this.

In addition, we have added the following references to our manuscript:

  • Sasikumar PG, Sudarshan NS, Adurthi S, Ramachandra RK, Samiulla DS, Lakshminarasimhan A, Ramanathan A, Chandrasekhar T, Dhudashiya AA, Talapati SR, Gowda N, Palakolanu S, Mani J, Srinivasrao B, Joseph D, Kumar N, Nair R, Atreya HS, Gowda N, Ramachandra M. PD-1 derived CA-170 is an oral immune checkpoint inhibitor that exhibits preclinical anti-tumor efficacy. Commun Biol. 2021 Jun 8;4(1):699. doi: 10.1038/s42003-021-02191-1. PMID: 34103659; PMCID: PMC8187357.
  • Guo L, Overholser J, Darby H, Ede NJ, Kaumaya PTP. A newly discovered PD-L1 B-cell epitope peptide vaccine (PDL1-Vaxx) exhibits potent immune responses and effective anti-tumor immunity in multiple syngeneic mice models and (synergizes) in combination with a dual HER-2 B-cell vaccine (B-Vaxx). Oncoimmunology. 2022 Oct 5;11(1):2127691. doi: 10.1080/2162402X.2022.2127691. PMID: 36211807; PMCID: PMC9542669.
  • Sasikumar PG, Sudarshan NS, Adurthi S, Ramachandra RK, Samiulla DS, Lakshminarasimhan A, Ramanathan A, Chandrasekhar T, Dhudashiya AA, Talapati SR, Gowda N, Palakolanu S, Mani J, Srinivasrao B, Joseph D, Kumar N, Nair R, Atreya HS, Gowda N, Ramachandra M. PD-1 derived CA-170 is an oral immune checkpoint inhibitor that exhibits preclinical anti-tumor efficacy. Commun Biol. 2021 Jun 8;4(1):699. doi: 10.1038/s42003-021-02191-1. PMID: 34103659; PMCID: PMC8187357.

Comment 2.16 Is there any related in-depth therapeutic data to show (pre-clinical)?

Response: In our project, we showed significant modulation of PD-L1 in niraparib treated triple negative breast cancer and trending differences in PD-L1 in radiation or paclitaxel treated tumors and corresponding changes in tumor volume. We performed immunohistochemistry against PD-L1 expression in paclitaxel, radiation and niraparib treated tumors; however, these tumors had increases in PD-L1 that were only trending towards significance.

Reviewer 3 Report

Thank you for submitting your paper on the development and optimization of [89Zr]Atezolizumab for PD-L1 imaging in TNBC models. Overall, the paper provides interesting insights into this area, but there are some concerns and questions that need to be addressed.

Firstly, the PET signal observed in the bone joints of the mice in figure 2B raises concerns about the stability of [89Zr]Atezolizumab in vivo. Therefore, we recommend conducting a serum stability study to determine the stability of the radiolabeled antibody in circulation. Additionally, please share the iTLC data for 89Zr.

Secondly, we noted that the in vitro stability tests were only 24 hours, which seems insufficient considering the 8-day imaging period in the animal studies. We recommend conducting a longer-term stability study to assess the long-term viability of the radiolabeled antibody.

Thirdly, in figure 2A, the comparable uptake of tumors in blocking and non-blocking groups raises questions about the specificity of [89Zr]Atezolizumab for PD-L1. Further investigation is required to confirm the specificity of the radiolabeled antibody.

Finally, we are curious about the influence of PDL-1 expression under combination therapy of drugs and radiation. It would be valuable to investigate whether PDL-1 expression is altered under these conditions and, if so, how this impacts the sensitivity to immunotherapy.

We appreciate your contribution to this area of research and look forward to seeing how these concerns are addressed in future revisions of your paper.

Author Response

Dear Editor and Reviewers,

Thank you for the opportunity to submit a revised manuscript and for the time you have committed to improving the study. We appreciate your time and effort reviewing our work and look forward to working together to strengthen this manuscript. We believe we have addressed all your comments and suggestions as indicated below and in the revised manuscript.

Reviewer #3:

Comment 3.01 Thank you for submitting your paper on the development and optimization of [89Zr]-Atezolizumab for PD-L1 imaging in TNBC models. Overall, the paper provides interesting insights into this area, but there are some concerns and questions that need to be addressed.

Response: Thank you for your time and effort into reviewing this manuscript. We appreciate your feedback and guidance for improving this manuscript.

Comment 3.02 Firstly, the PET signal observed in the bone joints of the mice in figure 2B raises concerns about the stability of [89Zr]-Atezolizumab in vivo. Therefore, we recommend conducting a serum stability study to determine the stability of the radiolabeled antibody in circulation. Additionally, please share the iTLC data for 89Zr.

Response: Thank you for your comments. In our studies with 89Zr-Atezolizumab, it is noted that 89Zr PET signal is observed in bone joints; however, joint uptake of 89Zr is commonly observed in 89Zr PET imaging [1-3]. We have amended the supplemental figures to include iTLC data for our tracer 89Zr-Atezolizumab.

Comment 3.03 Secondly, we noted that the in vitro stability tests were only 24 hours, which seems insufficient considering the 8-day imaging period in the animal studies. We recommend conducting a longer-term stability study to assess the long-term viability of the radiolabeled antibody.

Response: Thank you for addressing this. Thank you, we respectfully note that assays with similar chemistry of conjugation of antibodies to 89Zr have been well documented indicating long-term stability, including Massicano et al. and Krache et al [4, 5].

Comment 3.04 Thirdly, in figure 2A, the comparable uptake of tumors in blocking and non-blocking groups raises questions about the specificity of [89Zr]-Atezolizumab for PD-L1. Further investigation is required to confirm the specificity of the radiolabeled antibody.

Response: Thank you for pointing this out. It is possible that, due to inherent heterogeneity of PD-L1 expression, tumors may differ in baseline PD-L1 expression. To confirm the specificity of 89Zr-Atezolizumab and account for changes in 89Zr Atezolizumab blocking, a baseline PD-L1 PET scan would be optimal to include. We have expanded in the discussion how optimization of chemistry with shorter-isotopes may allow for more frequent monitoring of changes in PDL1 expression.

Comment 3.05 Finally, we are curious about the influence of PDL-1 expression under combination therapy of drugs and radiation. It would be valuable to investigate whether PDL-1 expression is altered under these conditions and, if so, how this impacts the sensitivity to immunotherapy.

Response: Thank you for your suggestions. We agree that studying changes in PD-L1 expression in response to a combination of chemotherapy and radiation therapy would be interesting, as single agent therapy is rarely given for TNBC clinically. While this is of significant interest, it is beyond the scope of this manuscript, however we have expanded the discussion section and have indicated that this will be critical next steps. Importantly, imaging provides the only opportunity to temporally understand how PDL1 changes during these combination therapies in evaluation of the whole 3D tumor volume.

References:

  1. Marquez, B.V., et al., Evaluation of (89)Zr-pertuzumab in Breast cancer xenografts. Mol Pharm, 2014. 11(11): p. 3988-95.
  2. Stone, L.D., et al., (89)Zr-panitumumab PET imaging for preoperative assessment of ameloblastoma in a PDX model. Sci Rep, 2022. 12(1): p. 19187.
  3. Benedetto, R., et al., (89)Zr-DFO-Cetuximab as a Molecular Imaging Agent to Identify Cetuximab Resistance in Head and Neck Squamous Cell Carcinoma. Cancer Biother Radiopharm, 2019. 34(5): p. 288-296.
  4. Krache, A., et al., Preclinical Pharmacokinetics and Dosimetry of an (89)Zr Labelled Anti-PDL1 in an Orthotopic Lung Cancer Murine Model. Front Med (Lausanne), 2021. 8: p. 741855.
  5. Massicano, A.V.F., et al., Imaging of HER2 with [(89)Zr]pertuzumab in Response to T-DM1 Therapy. Cancer Biother Radiopharm, 2019. 34(4): p. 209-217.

Round 2

Reviewer 1 Report

[89Zr]Atezolizumab-PET imaging reveals longitudinal alterations in PDL1 during therapy in TNBC preclinical models

Feedback on v2 – 26-04-2023

Dear Editor and Reviewers,

Thank you for the opportunity to submit a revised manuscript and for the time you have committed to improving the study. We appreciate your time and effort reviewing our work and look forward to working together to strengthen this manuscript. We believe we have addressed all your comments and suggestions as indicated below and in the revised manuscript.

Reviewer #1

Comment 1.01 The authors describe the chemistry development, in vitro characterization and in vivo characterization in mouse tumor models of 89Zr-atezolizumab. The aim of the paper was to study the potential of 89Zr-Atezolizumab to measure PD-L1 expression in triple negative breast cancer. They describe the biodistribution. They state that they demonstrate that the tumor SUVmean was significantly higher than control tissue (1.00 ± 0.16 SUVmean versus 0.39 ± 0.08 SUVmean; for tumor and muscle respectively; and that therapy with neraparib increases the tumor:muscle SUVmean at 1 and 4 days compared to the control group (2.069 ± 0.101 SUVmean versus 2.461 ± 0.320 SUVmean, for day 1, and 2.098 ± 0.276 SUVmean versus 2.591 ± 0.224 SUVmean, for day 4; P = <0.05; N=5), however, paclitaxel and radiation only induced a slight upregulation (P>0.05) of PD-L1 in the tumor.

Response: We thank you for the accurate summary of our work and believe we have addressed the comments which have further strengthened this manuscript.

Comment 1.02 There are several issues with the paper which make it not fit for publication as is.

The optimal conjugation molar ratio found was 1:20; optimized radiochemical purity 98.88% ± 1.3% and stability was >95% after 24 h at 2-8 ºC. Stability is only tested to 24hrs at 2-8oC, which in a patient situation would be too short as mAb take several days to distribute/optimal imaging timepoint 4-7 days after injection, and in the patient the temperature is 37oC. As imaging is done up to 8 days after injection, the authors have presumably tested these conditions in vitro as well. We ask to present the data. If this was not done, the authors need to explain why this would not be required.

Response: Thank you, we respectfully note that assays with similar chemistry of conjugation of antibodies to 89Zr have been well documented indicating long-term stability, including Massicano et al. and Krache et al [1, 2].

The added references include data from the [89Zr]DFO-anti-PDL1 and the [89Zr]pertuzumab. The included limitation describes the fact that more extensive RCP testing should be performed “Limitations of the study include the absence of extended-term evaluation of the radiotracer stability in human and mouse serum in vitro.”

Comment 1.03 In vitro characterization was done MDA MB 231 and HCC38 cells. The authors do not show/refer to publication regarding the expression of PD-L1 in these cell lines. Please add. They describe a high binding of [89Zr]-Atezolizumab for both cell lines, but only internalization MDA-MB-231 cells. Why? Same question regarding PD-L1 expression in the BCM 3936 PDX cells? Why use another model for the animal experiments?

Response: Thank you for your comments. We have included references regarding PD-L1 expression in MDA-MB-231 and HCC38 cell lines (listed below). Due to the heterogenous expression of PD-L1 within TNBC cell lines (MDA-MB-231 and HCC38), a cell line with high expression of PD-L1 (MDA-MB-231) was used for internalization studies, to allow for study of minor changes in PD-L1 expression. PDX tumors (BCM 3936) were enrolled into this experiment to examine whether our [89Zr]-Atezolizumab radiotracer can be used in more clinically relevant tissues. These tissues exhibit physiological heterogeneity of PD-L1 expression to better model representative of clinical PD-L1 expression in TNBC. We have amended the discussion section to discuss this change.

Please note that we have added the following references:

-Mittendorf EA, Philips AV, Meric-Bernstam F, Qiao N, Wu Y, Harrington S, Su X, Wang Y, Gonzalez-Angulo AM, Akcakanat A, Chawla A, Curran M, Hwu P, Sharma P, Litton JK, Molldrem JJ, Alatrash G. PD-L1 expression in triple-negative breast cancer. Cancer Immunol Res. 2014 Apr;2(4):361-70. doi: 10.1158/2326-6066.CIR-13-0127. Epub 2014 Jan 10. PMID: 24764583; PMCID: PMC4000553.

-Karasar P, Esendagli G. T helper responses are maintained by basal-like breast cancer cells and confer to immune modulation via upregulation of PD-1 ligands. Breast Cancer Res Treat. 2014 Jun;145(3):605-14. doi: 10.1007/s10549-014-2984-9. Epub 2014 May 10. PMID: 24816762.

This comment is addressed sufficiently.

Comment 1.04 The authors extrapolate 89Zr-atezolizumab directly to PD-L1 expression. The IHC they show is just one image of tumor staining. There are no data with autoradiography shown that PD-L1 expression correlates with tracer uptake. There are no IHC data shown how PD-L1 expression changes during therapy, as the authors claim that occurs. In the material and methods section it is not clear who has evaluated if the slides indeed contain tumor? What area ( tumor containing? Stroma?) is analysed for PD-L1 expression as it has been done with digitized analyses. The authors are invited to show these additional data.

Response: Quantitative IHC was conducted post imaging to provide secondary analysis of PD-L1 expression. The increases in PD-L1 expression in central slice analysis of paclitaxel, radiation, and niraparib treated tumors were only trending towards significance. Importantly, this points to the need for imaging of the entire 3D tumor volume to be incorporated when evaluating tissue, as a single slice does not fully represent the entire heterogenous tumor volume. The tumors in this experiment were engrafted and excised post imaging for biological validation. To this end, all tumors used in this experiment are cancerous and do not contain non-cancerous regions, and therefore the PDL1 expression is being quantified on all cells within the tumor microenvironment. The histology demonstrates the high expression of PDL1 on the cancer cells themselves.

 This comment is addressed sufficiently.

Comment 1.05 In figure 3 and figure 4 there really is no reasonable uptake of the tracer in the tumor lesions. This seems to be mirrored by a SUVmean of 1. With these data it can not be judged if the therapy indeed increases PD-L1 expression/89Zr atezolizumab uptake.

Response: Thank you for your comments. We have updated the images scale to show a more reasonable uptake of [89Zr]-Atezolizumab. Also, we respectfully note that the quantitative evaluation with SUV is able to distinguish differences that the qualitative images are  unable to decipher, indicating the importance of incorporating quantitative evaluation into clinical workflows in the future.

Scaling did improve visibility, but conclusion can still not be derived from image. Quantitative outcome does support the message. Not clear whether imaging is performed before or after intervention at day 0 (paclitaxel, niraparib, radiation)?  Note the large difference in SUVmean at day 0 for paclitaxel. Note the typo of ‘pLaclitaxel’ in figure 4.B.

Comment 1.06 There is no reference regarding which methods the authors calculate the dosimetry and extrapolate it to the human situation. Please add.

Response: Thank you for addressing this. We have modified the text to include a reference for this.

Please note that the references have been modified to include the following:

-Kelly MP, Makonnen S, Hickey C, Arnold TC, Giurleo JT, Tavaré R, Danton M, Granados C, Chatterjee I, Dudgeon D, Retter MW, Ma D, Olson WC, Thurston G, Kirshner JR. Preclinical PET imaging with the novel human antibody 89Zr-DFO-REGN3504 sensitively detects PD-L1 expression in tumors and normal tissues. J Immunother Cancer. 2021 Jan;9(1):e002025. doi: 10.1136/jitc-2020-002025. PMID: 33483343; PMCID: PMC7831708.

 This comment is addressed sufficiently.

Comment 1.07 The dosimetry analyses reveal a high effective dose of 0.773 mSv/MBq totaling 28,6 mSv for a 37MBq injection ( see subscript table, line 241. This is higher than expected for other 89Zr mAbs. What is the reason? This will hamper application. However, in the discussion, the authors state 0,49mSv/MBq (line 2910. Which one is correct? And how is it calculated?

Response: Thank you for your comment. We have updated the text with the correct number of 0.773 mSv/MBq. This effective dose is consistent with the range reported in the literature for Zr-89, which is typically between 20 and 40 mSv for 37-47 MBq of Zr-89.  Although most dosimetry studies for Zr-89 have been conducted in humans, there have been some preclinical studies in mice. For example, Kelly et al. estimated an effective dose of 23.01 mSv in humans for 89Zr-DFO-REGN3504 – an anti-PDL1 antibody - using animal models [3]. However, it is important to note that dosimetry values can differ between species and specific radiotracers. For instance, Lindenberg et al. reported an effective dose of 12.21 mSv for 89Zr-Panitumumab in humans [4], while Ulaner et al. estimated an effective dose of 20 mSv for 89Zr-Pertuzumab in humans [5]. We have modified the discussion to account for this information.

 This comment is addressed sufficiently.

Comment 1.08 The introduction and discussion avoids discussion of the results with this 89Zr-atezolizumab compared to the one used in the clinical trial. How does the production/characteristics of the 89Zr atezolizumab tracer here compares to the one described by Bensch et al?

Response: Thank you for your comments and the opportunity to discuss our results in context of Bensch et al. Our work expands on the work described by Bensch et al. by focusing on the specificity (through in vivo blocking assays), long term organ biodistribution, and modulation of PD-L1 expression in relation to paclitaxel, radiation and PARP inhibitor therapy.

 Is meant that the tracer has the same production / characteristics as the tracer described by Bensch et al.?

Comment 1.09 Monoblasts are precursors for macrophages and not for B-or T lymphocytes, Please correct.

Response: Thank you for this correction. We have amended the text to account for the expression of PD-L1 in macrophages.

 This comment is addressed sufficiently.

Comment 1.10 The figure legends do not match the figures presented. Please correct

Response: Thank you for your suggestion. We have updated the figure legends.

 This comment is addressed sufficiently.

Comment 1.11 The text is not clear regarding SUV mean as is vs ratio SUVmean tumor to SUVmean muscle. Why use SUVmean? Especially in such a poorly visible area? SUVpeak would be a more appropriate choice.

Response: Thank you for your suggestion. This study intended to assess the overall PDL1 expression in the tumor across a heterogeneous landscape therefore wanted to take the entire tumor volume into account, however we agree that SUVpeak would be a reasonable method to assess regions of highest PDL1 uptake and have included this in the discussion.

The explanation is clear. The addition in the discussion cannot be found.

Comment 1.12 Regarding the blocking study (Figure 1): the material and methods and the results of blocking effect are not described in text.

Response: Thank you for noticing that. We have expanded the methods section and included additional information in the results section to emphasize the results.

The explanation is clear. The addition in the discussion cannot be found.

Comment 1.13 [187] There is an increase in spleen uptake visible after blocking (fig 2A). The text describes a decrease. 

Response: Thank you for addressing this. We have amended the text to reflect an increase in spleen uptake after blocking. The interpretation of this phenomenon is that the blocking dose of atezolizumab increased [89Zr]-Atezolizumab passive blood presence, which is retained through spleen filtration. We would also point out that the increase in uptake in spleen was not significant.

Thank you for the correction. 

Comment 1.14 It seems there are other significant differences, but there is no asterix. Please clarify if you tested the differences for all tissues tested? If not, why not? What is the interpretation of the increase in the spleen?

 Response: Thank you for your observation. We tested the differences for all organs and have updated the figure to show the Asterix. The high uptake in the spleen probably occurs due to the saturation, by the blocking dose, of receptors in healthy tissues that express low levels of PD-L1, resulting in an increased quantity of radiotracer available in the bloodstream, as evidenced by its high blood uptake. This surplus radiotracer is then more readily available to bind to organs expressing higher levels of PD-L1, such as spleen, blood, heart, and skin.

This topic was addressed the following way:

By co-injecting cold antibody, the spleen, blood, heart, and skin, uptakes increase, while 344

the tumor uptake decreases (Figure 2A). One potential reasoning is that this occurs due to 345

the saturation, by the blocking dose, of receptors in healthy tissues that express low levels 346

of PD-L1, resulting in an increased quantity of radiotracer available in the bloodstream, 347

as evidenced by its high blood uptake. This surplus radiotracer is then more readily 348

available to bind to organs expressing higher levels of PD-L1. Although the biodistribu- 349

tion blocking study evinced a reduction in tumor uptake and a marginal increase in spleen 350

uptake, the imaging study showed the opposite trend to the biodistribution blocking 351

study (Figure 2B). Specifically, we observed a slight, but not significant, increase in tumor 352

SUVmean (P = 0.3388) in the blocking group, concomitant with a decrease in spleen SUVmean 353

(P = 0.0437). While these preliminary findings warrant confirmation through replication 354

in a larger cohort and validation via biodistribution studies, they align with previously 355

reported results [18, 38], including one conducted in non-human primates [39]. This ob- 356

servation suggests that PET imaging using [89Zr]-Atezolizumab can be performed in pa- 357

tients who are already receiving atezolizumab with no reduction in tumor uptake.

A few comments on this:

-       Also increase in pancreas uptake.

-       Why would blocking mainly take place in organs with low pd-l1 expression? Would the easily accessible (well perfused) organs with much pd-l1 uptake not be especially susceptible for atezolizumab uptake?

-       Increase in tumour uptake is not clear from this figure.

-       Figure 2C: without blocking dose (2.5 mg)?

Comment 1.15 What is the reason to use a PARP inhibitor, paclitaxel and radiation for increase in PD-L1 expression. Please add some information in the introductions

Response: Thank you for this relevant question. As TNBC lacks targetable treatment receptors, standard of care treatment options includes non-specific cytotoxic therapy, such as chemotherapy or radiation, and PARP inhibitors. Our goal was to evaluate the changes in PD-L1 expression by standard care therapy. This could be used to understand how PDL1 changes during combination therapies in order to most appropriately identify optimal sequencing and timing of treatments when given in combination with immunotherapy. While PARP is a novel target being explored in TNBC clinical trials, paclitaxel and radiation therapy are standard treatment options that may be used in clinical care of TNBC, along with immunotherapies. We have updated the introduction and discussion to explain this point better.

The explanation is clear. The addition in the discussion cannot be found.

Comment 1.16 The author Prof Lapi, is not correct in the authors list.

Response: Thank you for addressing this. We have corrected the author’s name and adjusted this in the manuscript.

Thank you for the correction.

Comment 1.17 [43] Specify immune cell interactions, better even leave out as it seems not be correct

Response: Thank you for your comment. At your suggestion, we have modified the text discussing immune cell interactions, which allows us to highlight the types of treatments used for TNBC.

 This comment is addressed sufficiently.

Comment 1.18 [49] atezolizumab has been retracted as therapy for TN breast cancer. Other alternative is pembrolizumab which is also indicated with PD-L1 expression

Response: Thank you for this comment. While atezolizumab may be retracted as a therapy for TNBC, 89Zr-Atezolizumab PET imaging can still allow for non-invasive imaging of the tumor immune microenvironment and predictive treatment response based on changes in PD-L1 expression. Importantly, this provides an opportunity to use an FDA-approved antibody for alternative purposes. Further, PDL1 expression is still evaluated in biopsy samples to indication of other immune checkpoint blockade therapies (one of which is pembrolizumab). Pembrolizumab, a different immunotherapy, targets PD-1 expression, which is primarily expressed in immune cells, rather than tumors cells, increasing the likelihood for background. We have adjusted the discussion to include this information.

 This comment is addressed sufficiently.

Comment 1.19 [54] and also the other way around: patients with low PD-L1 expressing may have a better response than expected from IHC

Response: Thank you for your comments. This is true and we have updated the introduction to include this information.  

 This comment is addressed sufficiently.

Comment 1.20 [70] please reframe the results of the paper of Bensch et al. There is only a significant trend reported for IHC by SP142 and SUVmax, but not for SP263. No statistical difference in mean was find between IHC0,1 or 2 and SUVmax for SP142.

Response: Thank you for your comments. In Bensch et al., SP142 references nonmalignant tissue. Due to the inherent heterogeneity of PD-L1 expression in tissue, it is possible that the biopsied region of tissue is not representative of PD-L1 expression within the entire tissue, which could be provided with PET imaging.

I don’t understand this. Why does SP142 refers nonmalignant tissues? And why is it then compared to tumour uptake? Unless there is a reason that only SP142 is relevant, the non- significant results of SP263 should be mentioned too.

Comment 1.21 [122] please describe why two different xenograft tumor models are used to study in vivo specificity+biodistribution and tumour uptake and why this could not have been done in the same mice.

Response: PDX tumors were enrolled into this experiment to examine whether our 89Zr-Atezolizumab radiotracer can image for physiological heterogeneity of PD-L1 expression in a model that most recapitulates clinical PD-L1 expression. This animal model is expensive, therefore we initially utilized a xenograft model with one cell line  to evaluate tumor uptake that has high expression of PD-L1 (MDA-MB-231). Once we achieved favorable results, we moved to PDX animal model. Dosimetry studies in humans are typically conducted in healthy individuals; thus, we performed the biodistribution for dosimetry calculation in healthy balb-c mice. The discussion has been modified to include the rational for using PDX model.

The explanation is clear, thank you. 

Comment 1.22 Which PD-L1 antibody did you use for IHC?

Response: Thank you for your comment. In supplementary methods, we have noted that an anti-human PD-L1 antibody from Abcam was used (ab210931, Abcam, Cambridge, MA).

The explanation is clear, thank you. 

Comment 1.23 [130] How many BCM 3936 PDX tumor bearing mice were injected?

Response: Thank you for your comment. For initial imaging studies, N = 5 BCM 3936 tumors were imaged with 89Zr-Atezolizumab PET imaging. For therapy studies, N = 4/group (total of 16) BCM 3936 tumors were enrolled into the experiment. These details can be found in the Supplemental Methods.

The explanation is clear, thank you. 

Comment 1.24 [144] please specify when the treatment occurred.

Response: Thank you for your suggestion. We have amended the methods section to include details regarding treatment schedule.

Thank you for adding more information. It is still not clear when the imaging occurs. Is this before or after treatment?

Comment 1.25 [148] clarify when the mice are euthanized and organs were collected.

Response: Thank you for your comment. We have adjusted the text to make it clearer.

Thank you for the additional information.

Comment 1.26 [157] was the data normally distributed?

Response: Thank you for your comments, the data in the histogram distribution in Figure 3B confirm that the data is normally distributed.

Thank you for the clarification.

Comment 1.27 [189] in the result section one should not discuss the results. Please move to the discussion

Response: We appreciate this suggestion and have updated the methods and discussion.

Thank you for the adjustment

Comment 1.28 206 it is nowhere shown that 89atezolizumab has a heterogeneous uptake in the tumor larger SD between lesions does not mean heterogeneity within one lesions. What is more, the very low levels of uptake will lead to larger SD due to measurement variability.

Response: Thank you for your comment. In figure 3B, we have shown a histogram distribution of 89Zr-Atezolizumab uptake within muscle and tumor regions. We show that tumor regions have a more heterogeneous distribution of 89Zr-Atezolizumab compared to normal/muscle tissue.

Thank you for the clarification.

Comment 1.29 235 with 89Zr mAb PET scans one does not acquire Time activity curves, but statis images at several time points. Please correct.

Response: Thank you, the text has been corrected.

Thank you for the adjustment

Comment 1.30 [figure 2] it is not clear when the blocking dose is provided and when the organ uptake is assessed.

Response: Thank you for your comment. The blocking group received 2.5 mg of unlabeled atezolizumab concurrently with [89Zr]-Atezolizumab  and where euthanized 7 days p.i followed by assessment of the organ uptake. The figure legend was updated.

Thank you for the adjustment

Comment 1.31 [figure 2C] to visualize an effect over time, a line graph is recommended.

Response: Thank you for your suggestion. To study the distribution of 89Zr-Atezolizumab in vivo, we used a biodistribution study at different timepoints to assess the localization of 89Zr-Atezolizumab in different organs. We respectfully believe a bar graph is able to show the wide range of distribution across different organs over the course of eight days. Using a line graph would make it more difficult to assess sensitive differences in organ accumulation.

Thank you for the clarification.

Comment 1.32 [figure 3] C is too small to properly evaluate.

Response: We apologize. We have expanded to include this in the Supplemental Material section this figure and other IHC figures with better resolution and larger size.

Thank you for the adjustment

Comment 1.33 [figure 3] Regarding of the SUV mean values used. Is there any difference in clearance and thus tracer supply between the animals expected?

Response: Thank you for addressing the difference between tracer clearance and supply. Non-cancerous cells, such as lungs, liver, kidney and pancreas, are known to express PD-L1, which can result in heterogeneity within the animal. To account to the clearance and decay, imaging quantification was normalized based on tracer decay at 7 days post injection of (1.85-3.7 MBq).

Thank you for the answer. Equal tracer supply is however still not checked/ assured.

Comment 1.34 [203] please specify, is the PD-L1 fraction for tumor cells only?

Response: We thank the reviewer for this question. PD-L1 fraction is used for the tumor only. A threshold was set based on mean muscle + 2 standard deviations and all pixels with an SUV above this threshold were deemed positive for PD-L1. Muscle was used as normal background due to normal expression of PDL1. This fraction is only applicable to the tumor.

Thank you for the clarification.

Comment 1.35 [269] spleen uptake increased instead of decreased.

Response: Thank you for addressing this. As we have previously noted, we have amended the text to reflect an increase in spleen uptake after blocking.

References:

  1. Krache, A., et al., Preclinical Pharmacokinetics and Dosimetry of an (89)Zr Labelled Anti-PDL1 in an Orthotopic Lung Cancer Murine Model. Front Med (Lausanne), 2021. 8: p. 741855.
  2. Massicano, A.V.F., et al., Imaging of HER2 with [(89)Zr]pertuzumab in Response to T-DM1 Therapy. Cancer Biother Radiopharm, 2019. 34(4): p. 209-217.
  3. Kelly, M.P., et al., Preclinical PET imaging with the novel human antibody (89)Zr-DFO-REGN3504 sensitively detects PD-L1 expression in tumors and normal tissues. J Immunother Cancer, 2021. 9(1).
  4. Lindenberg, L., et al., Dosimetry and first human experience with (89)Zr-panitumumab. Am J Nucl Med Mol Imaging, 2017. 7(4): p. 195-203.
  5. Ulaner, G.A., et al., First-in-Human Human Epidermal Growth Factor Receptor 2-Targeted Imaging Using (89)Zr-Pertuzumab PET/CT: Dosimetry and Clinical Application in Patients with Breast Cancer. J Nucl Med, 2018. 59(6): p. 900-906.

Author Response

Dear Editor and Reviewers,

Thank you for the opportunity to submit a revised manuscript and for the time you have committed to improving the study. We appreciate your time and effort reviewing our work and look forward to working together to strengthen this manuscript. We believe we have addressed all your comments and suggestions as indicated below and in the revised manuscript.

Reviewer #1

Comment 1: The added references include data from the [89Zr]DFO-anti-PDL1 and the [89Zr]pertuzumab. The included limitation describes the fact that more extensive RCP testing should be performed “Limitations of the study include the absence of extended-term evaluation of the radiotracer stability in human and mouse serum in vitro.”

Response 1: Thank you for your feedback and addressing this limitation of the study. We agree that a limitation of this study is the lack of longitudinal stability data in vitro and in vivo; however given the allotted five day window of response to reviewers, we are currently unable to assess the stability of our tracer experimentally. In Massicano et al., tracer stability was assessed until day 7 and, while there is a reduction in the tracer stability, the majority of the tracer remained stable.

We have revised the discussion to include the following text:

“Importantly, the long-term stability of similar radiolabeled antibodies has been previously demonstrated [1, 2]; however, longitudinal stability of [89Zr]-Atezolizumab should be assessed preclinically and clinically through iTLC.”

Comment 2: Scaling did improve visibility, but conclusion can still not be derived from image. Quantitative outcome does support the message. Not clear whether imaging is performed before or after intervention at day 0 (paclitaxel, niraparib, radiation)?  Note the large difference in SUVmean at day 0 for paclitaxel. Note the typo of ‘pLaclitaxel’ in figure 4.B.

Response 2: Thank you for acknowledging the improved quality of the figure and your inquiry regarding the scheduling of treatment and imaging. We apologize for our typo in Figure 4B and we have adjusted the figure legend for figure 4B to address this typo. [89Zr]-Atezolizumab occurred on days 0, 1 and 4. Paclitaxel therapy occurred on days 0 and 3. Radiation therapy occurred on days 0, 1, 2, and 3. Niraparib therapy occurred on days 0, 1, 2, and 3. Baseline imaging was conducted prior to start of treatment so we can monitor intratumoral changes in PD-L1 expression in response to therapy. This has been clarified in the methods section.

The discussion has been adjusted to include the following text:

“[89Zr]-Atezolizumab therapy studies have been conducted prior and following paclitaxel, niraparib or radiation therapy to study the dynamics of PD-L1 expression in TNBC tumors.”

Comment 3: Is meant that the tracer has the same production / characteristics as the tracer described by Bensch et al.?

Response 3: Thank you for allowing us the opportunity to compare our radiotracer production to Bensch et al. Our [89Zr]-Atezolizumab makes use of a different chelator (DFO-Bz-NCS), whereas Bensch uses N-succinyldesferrioxamine. Moreover, we present a range of specific activity of [89Zr] to label Atezolizumab (15-74 MBq/mg compared to 37 MBq/mg in Bensch et al.,) to optimize labeling conditions.

Comment 4: This topic was addressed the following way:

By co-injecting cold antibody, the spleen, blood, heart, and skin, uptakes increase, while 344

the tumor uptake decreases (Figure 2A). One potential reasoning is that this occurs due to 345

the saturation, by the blocking dose, of receptors in healthy tissues that express low levels 346

of PD-L1, resulting in an increased quantity of radiotracer available in the bloodstream, 347

as evidenced by its high blood uptake. This surplus radiotracer is then more readily 348

available to bind to organs expressing higher levels of PD-L1. Although the biodistribu- 349

tion blocking study evinced a reduction in tumor uptake and a marginal increase in spleen 350

uptake, the imaging study showed the opposite trend to the biodistribution blocking 351

study (Figure 2B). Specifically, we observed a slight, but not significant, increase in tumor 352

SUVmean (P = 0.3388) in the blocking group, concomitant with a decrease in spleen SUVmean 353

(P = 0.0437). While these preliminary findings warrant confirmation through replication 354

in a larger cohort and validation via biodistribution studies, they align with previously 355

reported results [18, 38], including one conducted in non-human primates [39]. This ob- 356

servation suggests that PET imaging using [89Zr]-Atezolizumab can be performed in pa- 357

tients who are already receiving atezolizumab with no reduction in tumor uptake.

A few comments on this:

-       Also increase in pancreas uptake.

-       Why would blocking mainly take place in organs with low pd-l1 expression? Would the easily accessible (well perfused) organs with much pd-l1 uptake not be especially susceptible for atezolizumab uptake?

-       Increase in tumour uptake is not clear from this figure.

-       Figure 2C: without blocking dose (2.5 mg)?

Response 4: Significant uptakes in [89Zr]-Atezolizumab uptake is observed in blood, heart, pancreas, skin and tumor in mice that received a blocking dose prior to [89Zr]-Atezolizumab imaging. In blocking conditions, because tissue with high expression of PD-L1 is blocked by the primary dose of cold Atezolizumab, it is expected that the residual dose of [89Zr]-Atezolizumab would accumulate in regions of lower PD-L1 expression until finally secreted. The comment regarding figure 2C is correct. A blocking dose was not used in figure 2C because we were seeking to study the temporal kinetics of tracer uptake over the course of 8 days within organs of interest. This has been clarified in the caption.

Comment 5: I don’t understand this. Why does SP142 refers nonmalignant tissues? And why is it then compared to tumour uptake? Unless there is a reason that only SP142 is relevant, the non- significant results of SP263 should be mentioned too.

Response 5: Thank you for the opportunity to address this. Per Bensch et al., SP142 refers to examples of three normal non-malignant lymph nodes, which are presented to show heterogeneous expression of PD-L1 staining in healthy tissue. We believe this was presented to show biological validation of [89Zr]-Atezolizumab uptake in tissue that is not specifically cancerous.

Please note that the text has been adjusted to include the following text:

“Bensch found correlation between non-malignant lymph node tissue and [89Zr]-Atezolizumab uptake.”

Comment 6: Thank you for adding more information. It is still not clear when the imaging occurs. Is this before or after treatment?

Response 6: Thank you for your inquiry regarding the scheduling of treatment and imaging. [89Zr]-Atezolizumab occurred on days 0, 1 and 4. Paclitaxel therapy occurred on days 0 and 3. Radiation therapy occurred on days 0, 1, 2, and 3. Niraparib therapy occurred on days 0, 1, 2, and 3. Baseline imaging was conducted prior to start of treatment so we can monitor intratumoral changes in PD-L1 expression in response to therapy.

The discussion has been adjusted to include the following text:

“[89Zr]-Atezolizumab therapy studies have been conducted prior and following paclitaxel, niraparib or radiation therapy to study the dynamics of PD-L1 expression in TNBC tumors.”

Comment 7: Thank you for the answer. Equal tracer supply is however still not checked/ assured.

Response 7: Thank you for your comments regarding tracer supply. For in vivo imaging studies, we consistently administered 50 µCi of [89Zr]-Atezolizumab. [89Zr]-Atezolizumab is labeled at a consistent specific activity, resulting in an administered dose of 12 µg injected per mouse. No additional validation methods were used.

Reviewer 2 Report

Thank you for your updates. Please check the format seriously and be consist throughout the manuscript. Good luck.

Author Response

Dear Editor and Reviewers,

Thank you for the opportunity to submit a revised manuscript and for the time you have committed to improving the study. We appreciate your time and effort reviewing our work and look forward to working together to strengthen this manuscript. We believe we have addressed all your comments and suggestions as indicated below and in the revised manuscript.

Reviewer #2:

Thank you for your updates. Please check the format seriously and be consist throughout the manuscript. Good luck.

Round 2 Response: Thank you for your comments and your continued dedication towards strengthening this manuscript.